# POISON-SPLAT: COMPUTATION COST ATTACK ON 3D GAUSSIAN SPLATTING

**Jiahao Lu[1][*][†]   Yifan Zhang[2][*]   Qiuhong Shen[1]   Xinchao Wang[1][‡]   Shuicheng Yan[2,1][‡]**
[1] National University of Singapore    [2] Skywork AI
`jiahao.lu@u.nus.edu`, `yifan.zhang7@kunlun-inc.com`, `qiuhong.shen@u.nus.edu`,
`xinchao@nus.edu.sg`, `shuicheng.yan@kunlun-inc.com`

## ABSTRACT

3D Gaussian splatting (3DGS), known for its groundbreaking performance and efficiency, has become a dominant 3D representation and brought progress to many 3D vision tasks. However, in this work, we reveal a significant security vulnerability that has been largely overlooked in 3DGS: **the computation cost of training 3DGS could be maliciously tampered by poisoning the input data**. By developing an attack named *Poison-splat*, we reveal a novel attack surface where the adversary can poison the input images to **drastically increase the computation memory and time** needed for 3DGS training, pushing the algorithm towards its worst computation complexity. In extreme cases, the attack can even consume all allocable memory, leading to a Denial-of-Service (DoS) that disrupts servers, resulting in practical damages to real-world 3DGS service vendors. Such a computation cost attack is achieved by addressing a bi-level optimization problem through three tailored strategies: attack objective approximation, proxy model rendering, and optional constrained optimization. These strategies not only ensure the effectiveness of our attack but also make it difficult to defend with simple defensive measures. We hope the revelation of this novel attack surface can spark attention to this crucial yet overlooked vulnerability of 3DGS systems.[1]

## 1 INTRODUCTION

3D reconstruction, aiming to build 3D representations from multi-view 2D images, holds a critical position in computer vision and machine learning, extending its benefits across broad domains including entertainment, healthcare, archaeology, and the manufacturing industry. A notable example is Google Map with a novel feature for NeRF-based (Mildenhall et al., 2021) 3D outdoor scene reconstruction (Google, 2023). Similarly, commercial companies like LumaAI, KIRI, Polycam and Spline offer paid service for users to generate 3D captures from user-uploaded images or videos. Among these popular 3D applications, many are powered by a novel and sought-after 3D reconstruction paradigm known as 3D Gaussian Splatting (3DGS).

3DGS, introduced by Kerbl et al. (2023) in 2023, has quickly revolutionized the field of 3D Vision and achieved overwhelming popularity (Chen & Wang, 2024; Fei et al., 2024; Wu et al., 2024b; Dalal et al., 2024). 3DGS is not powered by neural networks, distinguishing itself from NeRF (Mildenhall et al., 2021), the former dominant force in 3D vision. Instead, it captures 3D scenes by learning a cloud of 3D Gaussians as an explicit 3D model and uses rasterization to render multiple objects simultaneously. This enables 3DGS to achieve significant advantages in rendering speed, photo-realism, and interpretability, making it a game changer in the field.

One intriguing property of Gaussian Splatting is its **flexibility in model complexity**. Unlike NeRF or any other neural-network based algorithms which have a pre-determined and fixed computational complexity based on network hyper-parameters, 3DGS can dynamically adjust its complexity according to the input data. During the 3DGS training process, the size of learnable parameters, *i.e.* the number of 3D Gaussians is dynamically adjusted to align with the complexity of the underlying 3D scene. More specifically, 3DGS algorithm uses an *adaptive density control* strategy to flexibly

---

*Authors contributed equally. [†] This work was performed when Jiahao Lu was an intern at Skywork AI.
[‡]Corresponding authors
[1]Our code is available at `https://github.com/jiahaolu97/poison-splat`.

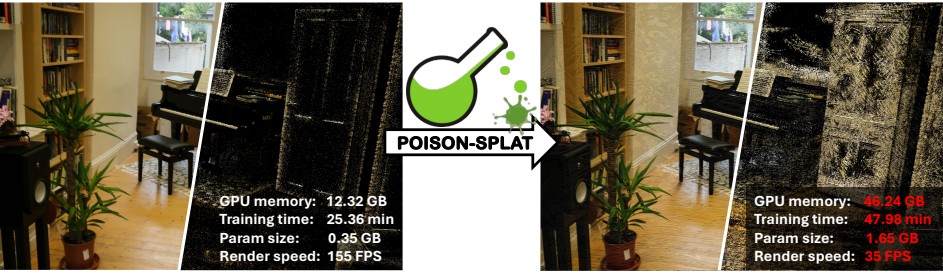

Figure 1: Our *Poison-splat* attack adds perturbations to input images, making 3D Gaussian Splatting need significantly more parameters to reconstruct the 3D scene, leading to huge increases in GPU memory consumption, training time and rendering latency. Here, we visualize the input image and the underlying 3D Gaussians of a clean view (left) and its corresponding poisoned view (right).

increase or decrease the number of Gaussians for optimizing reconstruction, leading to variable computational complexity in terms of GPU memory occupancy and training time cost.

The above flexibility in design aims to provide training advantages. However, this flexibility can also become a vulnerability. In this paper, we reveal a severe yet unnoticed attack vector: the flexibility of 3DGS complexity can be abused to over-consume computation resources such as GPU memory and significantly drag down training speed of Gaussian Splatting system, pushing the training process to its worst computational complexity.

We develop a computation cost attack method *Poison-splat* as a proof-of-concept of this novel attack vector. *Poison-splat* takes the form of training data poisoning (Tian et al., 2022), where the attacker manipulates the input data fed into the victim 3DGS system. This is practical in real-world scenarios, as commercial 3D service providers like Kiri (KIRI), Polycam (Polycam) and Spline (Spline) receive images or videos from end users to generate 3D captures. An attacker can hide themselves among normal users to submit poisoned data and launch the attack stealthily, or even secretly manipulate data uploaded by other users. During peak usage periods, this attack competes for computing resources with legitimate users, slowing down service response times and potentially causing severe consequences like service crashes, leading to significant financial losses for service providers.

We formulate *Poison-splat* attack as a max-min problem. The inner optimization is the learning process of 3D Gaussian Splatting, *i.e.*, minimizing the reconstruction loss given a set of input images and camera poses, while the outer optimization problem is to maximize the computation cost of solving the inner problem. Although accurately solving this bi-level optimization is often infeasible, we find that the attacker can use a *proxy model* to approximate the inner minimization and focus on optimizing the outer maximization objective. Moreover, we observe an important relationship that memory consumption and rendering latency exhibit a clear positive correlation with the number of 3D Gaussians in training. Therefore, the attacker can use the number of Gaussians in proxy model training as a metric to depict computation costs for outer optimization. Motivated by these insights, our *Poison-splat* attack uses image total variation loss as a prior to guide the over-densification of 3D Gaussians, and can approximately solve this bi-level optimization problem in a cheap way.

The contributions of this work can be concluded in three-folds:

- We reveal that the flexibility in model complexity of 3DGS can become a security backdoor, making it vulnerable to computation cost attack. This vulnerability has been largely overlooked by the 3D vision and machine learning communities. Our research shows that such attacks are feasible, potentially causing severe financial losses to 3D service providers.

- We formulate the attack on 3D Gaussian Splatting as a data poisoning attack problem. To our best knowledge, there is no previous work investigating how to poison training data to increase the computation cost of machine learning systems.

- We propose a novel attack algorithm, named *Poison-splat*, which significantly increases GPU memory consumption and slows down training procedure of 3DGS. We hope the community can recognize this vulnerability and develop more robust 3D Gaussian Splatting algorithms or defensive methods to mitigate such attacks.

## 2 PRELIMINARIES

In this section, we provide essential backgrounds on 3D Gaussian Splatting and resource-targeting attacks, which are crucial for understanding our attack method proposed in Section 3. Due to page constraints, we put related works in Appendix Section A.

### 2.1 3D GAUSSIAN SPLATTING

3D Gaussian Splatting (3DGS) (Kerbl et al., 2023) has revolutionized the 3D vision community with its superior performance and efficiency. It models a 3D scene by learning a set of 3D Gaussians $\mathcal{G} = \{G_i\}$ from multi-view images $\mathcal{D} = \{V_k, P_k\}_{k=1}^N$, where each view consists of a ground-truth image $V_k$ and the corresponding camera poses $P_k$. We denote $N$ as the total number of image-pose pairs in the training data. As for the model, each 3D Gaussian $G_i = \{\mu_i, s_i, \alpha_i, q_i, c_i\}$ has 5 optimizable parameters: center coordinates $\mu_i$, scales $s_i$, opacity $\alpha_i$, rotation quaternion $q_i$, and associated view-dependent color represented as spherical harmonics $c_i$.

To render $\mathcal{G}$ from 3D Gaussian space to 2D images, each Gaussian is first projected into the camera coordinate frame given a camera pose $P_k$ to determine the depth of each Gaussian, which is the distance of that Gaussian to the camera view. The calculated depth is used for sorting Gaussians from near to far, and then the colors in 2D image are rendered in parallel by alpha blending according to the depth order of adjacent 3D Gaussians. We denote the 3DGS renderer as a function $\mathcal{R}$, which takes as input the Gaussians $\mathcal{G}$ and a camera pose $P_k$ to render an image $V_k' = \mathcal{R}(\mathcal{G}, P_k)$.

The training is done by optimizing the reconstruction loss between the rendered image and ground truth image for each camera view, where the reconstruction loss is a combination of $\mathcal{L}_1$ loss and structural similarity index measure (SSIM) loss $\mathcal{L}_{\text{D-SSIM}}$ (Brunet et al., 2011):

$$\min_{\mathcal{G}} \mathcal{L}(\mathcal{D}) = \min_{\mathcal{G}} \sum_{k=1}^N \mathcal{L}(V_k', V_k) = \min_{\mathcal{G}} \sum_{k=1}^N (1 - \lambda)\mathcal{L}_1(V_k', V_k) + \lambda \mathcal{L}_{\text{D-SSIM}}(V_k', V_k), \quad (1)$$

where $\lambda$ is a trade-off parameter. To ensure finer reconstruction, 3DGS deploys *adaptive density control* to automatically add or remove Gaussians from optimization. Adding Gaussians is called densification, while removing Gaussians is called pruning. Densification is performed when the view-space positional gradient $\nabla_{\mu_i}\mathcal{L}$ is larger than a pre-set gradient threshold $\tau_g$ (by default 0.0002). Pruning occurs when the opacity $\alpha_i$ falls below the opacity threshold $\tau_\alpha$. Both densification and pruning are non-differentiable operations. A more formal and comprehensive introduction is provided in Appendix Section B.

### 2.2 RESOURCE-TARGETING ATTACK

A similar concept in computer security domain is the Denial-of-Service (DoS) attack (Elleithy et al., 2005; Aldhyani & Alkahtani, 2023; Bhatia et al., 2018). A Dos attack aims to overwhelmly consume a system's resources or network (Mirkovic & Reiher, 2004; Long & Thomas, 2001), making it unable to serve legitimate users. Common methods include flooding the system with excessive requests or triggering a crash through malicious inputs. Such attacks pose severe risks to real-world service providers, potentially resulting in extensive operational disruptions and financial losses. For example, Midjourney, a generative AI platform, experienced a significant 24-hour system outage (Akash, 2024), potentially caused by employees from another generative AI company who were allegedly scraping data, causing a Denial-of-Service.

In the machine learning community, similar concepts are rarely mentioned. This might stem from the fact that once hyper-parameters are set in most machine learning models, such as deep neural networks, their computation complexity remains fixed. Regardless of input data content, most machine learning algorithms incur nearly constant computational costs and consume almost consistent computational resources. Despite this, only a few studies have focused on **resource-targeting attacks at inference stage** of machine learning systems. For example, Shumailov et al. (2021) first identified samples that trigger excessive neuron activations which maximise energy consumption and latency. Following works investigated other inference-stage attacks against dynamic neural networks (Hong et al., 2020; Chen et al., 2023) and language models (Chen et al., 2022b;a; Gao et al., 2023). However, to our knowledge, no previous works have targeted at attacking the training-stage computation cost of machine learning systems. Our work is pioneering in bridging this research gap through Gaussian Splatting, which features adaptively flexible computation complexity.

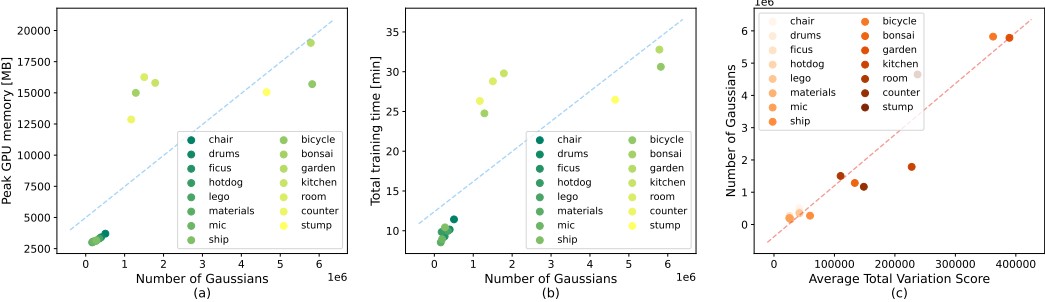

Figure 2: The figure illustrates the strong positive correlation between the number of Gaussians and (a) GPU memory occupancy, and (b) training time costs. Panel (c) shows the relationship between image sharpness, measured by the average total variation score, and the number of Gaussians.

## 3 POISON-SPLAT ATTACK

In this section, we first formulate the computation cost attack on 3D Gaussian Splatting in Section 3.1. Subsequently, we introduce a novel attack method in Section 3.2.

### 3.1 PROBLEM FORMULATION

We formulate our attack under data poisoning framework as follows. The victims are the service providers of 3D Gaussian Splatting (3DGS), who typically train their 3DGS models using a dataset of multi-view images and camera poses $\mathcal{D} = \{V_k, P_k\}_{k=1}^N$. We assume the attackers have the capability to introduce poisoned data within the entire dataset. This assumption is realistic as an attacker could masquerade as a regular user, submitting poisoned data covertly, or even covertly alter data submitted by legitimate users. This allows the attack to be launched stealthily and enhances its potential impact on the training process of 3DGS. We next detail the formulation of the attacker and victim.

**Attacker**. The attacker begins with the clean data $\mathcal{D} = \{V_k, P_k\}_{k=1}^N$ as input, manipulating it to produce poisoned training data $\mathcal{D}_p = \{\tilde{V}_k, P_k\}_{k=1}^N$, where the attacker does not modify the camera pose configuration files. Each poisoned image $\tilde{V}_k$ is perturbed from the original clean image $V_k$, aiming to maximize the computational costs of victim training, such as GPU memory usage and training time. The ultimate goal of the attacker is to significantly increase the computational resources, potentially leading to a denial-of-service attack by overwhelming the training system.

**Victim**. On the other hand, the victim receives this poisoned dataset $\mathcal{D}_p = \{\tilde{V}_k, P_k\}_{k=1}^N$ from the attacker, unaware that it has been poisoned. With this data, the victim seeks to train a Gaussian Splatting Model $\mathcal{G}$, striving to minimize the reconstruction loss (cf. Eq. (1)). The victim's objective is to achieve the lowest loss as possible, thus ensuring the quality of the Gaussian Splatting model.

**Optimization problem.** To summarize, the computation cost attack for the attacker can be formulated as the following max-min bi-level optimization problem:

$$\mathcal{D}_p = \arg\max_{\mathcal{D}_p} \mathcal{C}(\mathcal{G}^*), \quad \text{s.t.} \quad \mathcal{G}^* = \arg\min_{\mathcal{G}} \mathcal{L}(\mathcal{D}_p), \tag{2}$$

where the computation cost metric $\mathcal{C}(\mathcal{G})$ is flexible and can be designed by the attacker.

### 3.2 PROPOSED METHOD

To conduct the attack, directly addressing the above optimization is infeasible as the computation cost is not differentiable. To address this, we seek to find an approximation for the objective.

**Using the number of Gaussians as an approximation**. As is mentioned in Section 1, a major advantage of 3DGS is its flexibility to dynamically adjust model complexity (*i.e.*, the number of Gaussians) according to the complexity of the input data. This adaptability enhances the model's efficiency and fidelity in rendering complex scenes. However, this feature may also act as a double-edged sword, providing a potential backdoor for attacks. To explore this, we analyze how the number of Gaussians affects computational costs, including memory consumption and training time. Our findings, depicted in Figures 2(a-b), reveal a clear positive correlation between computational costs

---

**Algorithm 1** Poison-splat

---

**Input:** Clean dataset: $\mathcal{D} = \{V_k, P_k\}$; Perturbation range: $\epsilon$; Perturbation step size: $\eta$;
         The iteration number of inner optimization: $T$; The iteration number of outer optimization: $\tilde{T}$.
**Output:** Poisoned dataset: $\mathcal{D}_p = \{\tilde{V}_k, P_k\}$
 1: Initialize poisoned dataset: $\mathcal{D}_p = \mathcal{D}$
 2: Train a Gaussian Splatting proxy model $\mathcal{G}_p$ on clean dataset $\mathcal{D}$                                ▷ Eq. (1)
 3: **For** $t = 1, .., T$ **do**
 4:      Sample a random camera view $k$
 5:      $V'_k = \mathcal{R}(\mathcal{G}_p, P_k)$                              ▷ Get the rendered image under the camera view $k$
 6:      $\tilde{V}_k = V'_k$                                   ▷ Optimization of target image starts from rendered image
 7:      **For** $\tilde{t} = 1, .., \tilde{T}$ **do**
 8:          $\tilde{V}_k = \tilde{V}_k + \eta \cdot sign(\nabla_{\tilde{V}_k} \mathcal{S}_{TV}(\tilde{V}_k))$                              ▷ Eq. (4)
 9:          $\tilde{V}_k = \mathcal{P}_\epsilon(\tilde{V}_k, V_k)$                            ▷ Project into $\epsilon$-ball, *i.e.*, Eq. (5)
10:      **end While**
11:      Update proxy model $\mathcal{G}_p$ by minimizing $\mathcal{L}(V'_k, \tilde{V}_k)$ under current view          ▷ Eq. (1)
12:      Update poisoned dataset: $\mathcal{D}_p^k := (\tilde{V}_k, P_k)$          ▷ Update the $k$-th element of $\mathcal{D}_p$
13: **end While**

---

and the number of Gaussians used in rendering. In light of this insight, it is intuitive to use the number of Gaussians $\|\mathcal{G}\|$ to approximate the computation cost function involved in the inner optimization:

$$\mathcal{C}(\mathcal{G}) := \|\mathcal{G}\|. \tag{3}$$

**Maximizing the number of Gaussians by sharpening 3D objects**. Even with the above approximation, solving the optimization problem is still hard since the Gaussian densification operation in 3DGS is not differentiable. Hence, it would be infeasible for the attacker to use gradient-based methods to optimize the number of Gaussians. To circumvent this, we explore a strategic alternative. As shown in Figure 2(c), we find that 3DGS tends to assign more Gaussians to those objects with more complex structures and non-smooth textures, as quantified by the total variation score—a metric assessing image sharpness (Rudin et al., 1992). Intuitively, the less smooth the surface of 3D objects is, the more Gaussians the model needs to recover all the details from its 2D image projections. Hence, non-smoothness can be a good descriptor of complexity of Gaussians, *i.e.*, $\|\mathcal{G}\| \propto \mathcal{S}_{TV}(\mathcal{D})$. Inspired by this, we propose to maximize the computation cost by optimizing the total variation score of rendered images $\mathcal{S}_{TV}(\tilde{V}_k)$:

$$\mathcal{C}(\mathcal{G}) := \mathcal{S}_{TV}(\tilde{V}_k) = \sum_{i,j} \sqrt{\left|\tilde{V}_k^{i+1,j} - \tilde{V}_k^{i,j}\right|^2 + \left|\tilde{V}_k^{i,j+1} - \tilde{V}_k^{i,j}\right|^2}. \tag{4}$$

**Balancing attack strength and stealthiness via optional constrained optimization**. The above strategies enable the attack to significantly increase computational costs . However, this may induce an unlimited alteration to the image and result in a loss of semantic integrity in the generated view images (cf. Figure 4(b)), making it obvious to detect. Considering the strategic objectives of an adversary who may wish to preserve image semantics and launch the attack by stealth, we introduce an optional constrained optimization strategy. Inspired by adversarial attacks (Szegedy et al., 2013; Madry et al., 2018), we enforce a $\epsilon$-ball constraint of the $L_\infty$ norm on the perturbations:

$$\tilde{V}_k \in \mathcal{P}_\epsilon(\tilde{V}_k, V_k). \tag{5}$$

where $\mathcal{P}_\epsilon(\cdot)$ means to clamp the rendered poisoned image to be within the $\epsilon$-ball of the $L_\infty$ norm around the original clean image, *i.e.*, $\|\tilde{V}_k - V_k\|_\infty \leq \epsilon$. This constraint limits each pixel's perturbation to a maximum of $\epsilon$. By tuning $\epsilon$, an attacker can balance the destructiveness and the stealthiness of the attack, allowing for strategic adjustments for the desired outcome. If $\epsilon$ is set to $\infty$, the constraint is effectively removed, returning the optimization to its original, unconstrained form.

**Ensuring multi-view image consistency via proxy model**. An interesting insight from our study is that simply optimizing perturbations for each view image independently by maximizing total variation score does not effectively bolster attacks. As demonstrated in Figure 3(b), this image-level

TV-maximization attack is significantly less effective compared to our *Poison-splat* strategy. This mainly stems from the image-level optimization creating inconsistencies across the different views of poisoned images, undermining the attack's overall effectiveness.

Our solution is inspired by the view-consistent properties of the 3DGS model's rendering function, which effectively maintains consistency across multi-view images generated from 3D Gaussian space. In light of this, we propose to train a proxy 3DGS model to generate poisoned data. In each iteration, the attacker projects the current proxy model onto a camera pose, obtaining a rendered image $V'_k$. This image then serves as a starting point for optimization, where the attacker searches for a target $\tilde{V}_k$ that maximizes the total variation score within the $\epsilon$-bound of the clean image $V_k$. Following this, the attacker updates the proxy model with a single optimization step to mimic the victim's behavior. In subsequent iterations, the generation of poisoned images initiates with the rendering output from the updated proxy model. In this way, the attacker approximately solves this bi-level optimization by iteratively unrolling the outer and inner optimization, enhancing the attack's effectiveness while maintaining consistency across views. We summarize the pipeline of Poison-splat in Algorithm 1.

## 4 EXPERIMENTS

**Datasets**. All experiments in this paper are carried out on three common 3D datasets: (1) NeRF-Synthetic[2] (Mildenhall et al., 2021) is a synthetic 3D object dataset, which contains multi-view renderings of synthetic 3D objects. (2) Mip-NeRF360[3] dataset (Barron et al., 2022) is a 3D scene dataset, which is composed of photographs of large real-world outdoor scenes. (3) Tanks-and-Temples[4] dataset (Knapitsch et al., 2017) contains realistic 3D captures including both outdoor scenes and indoor environments.

**Implementation details**. In our experiments, we use the official implementation of 3D Gaussian Splatting[5] (Kerbl et al., 2023) to implement the victim behavior. Following their original implementation, we use the recommended default hyper-parameters, which were proved effective across a broad spectrum of scenes from various datasets. For testing black-box attack effectiveness, we use the official implementation of Scaffold-GS[6] (Lu et al., 2024b) as a black box victim. All experiments reported in the paper were carried out on a single NVIDIA A800-SXM4-80G GPU.

**Evaluation metrics**. We use the number of 3D Gaussians, GPU memory occupancy and the training time cost as metrics to evaluate the computational cost of 3DGS. A more successful attack is characterized by larger increases in the number of Gaussians, higher GPU memory consumption, and extended training time, compared with training on clean data. We also report the rendering speed of the resulted 3DGS models to show the increase in model complexity and degradation of inference latency as a byproduct of our attack.

### 4.1 MAIN RESULTS

We report the comparisons of the number of Gaussians, peak GPU memory, training time and rendering speed between clean and poisoned data across three different datasets in Tables 1 to show the effectiveness of our attack. The full results containing all scenes are listed in the Appendix Table 3 and Table 4. We provide visualizations of example poisoned datasets and corresponding victim reconstructions in Appendix Section F.

We want to highlight some observations from the results. First, our *Poison-splat* attack demonstrates the ability to craft a huge extra computational burden across multiple datasets. Even with perturbations constrained within a small range in an $\epsilon$-constrained attack, the peak GPU memory can be increased to over 2 times, making the overall maximum GPU occupancy higher than 24 GB - in the real world, this may mean that our attack may require more allocable resources than common GPU stations can provide, *e.g.*, RTX 3090, RTX 4090 and A5000. Furthermore, as we visualized the computational cost in Figure 3(a), the attack not only significantly increases the memory usage, but also greatly slows

---

[2]Dataset publicly accessible at https://github.com/bmild/nerf.

[3]Dataset publicly accessible at https://jonbarron.info/mipnerf360/

[4]Dataset publicly accessible at https://www.tanksandtemples.org/download/

[5]https://github.com/graphdeco-inria/gaussian-splatting

[6]https://github.com/city-super/Scaffold-GS

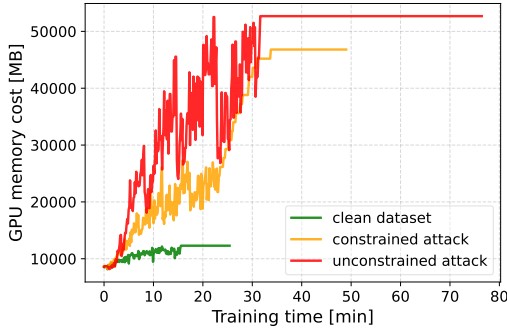 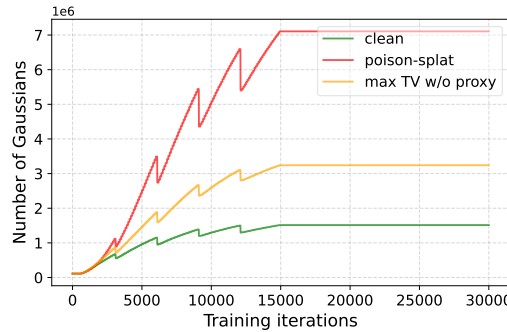

(a) Overall comparisons of computation cost in training time and GPU memory occupancy on MIP-NeRF360 (room).

(b) Number of Gaussians needed for reconstruction under our attack, naive TV-maximization attack and on the clean data of MIP-NeRF360 (room).

Figure 3: Figure (a) shows that our attack leads to a significant increase on the GPU memory and training time. Figure (b) shows that attacks simply maximizing total variation score at the image level are less effective compared to *Poison-splat*, which highlights the crucial role of the proxy model in our attack design.

down training speed. This property would further strengthen the attack, since the overwhelming GPU occupancy will last longer than normal training may take, making the overall loss of computation power higher. As shown in the bottom of Table 1, if we do not put constraints on the attacker and let it modify the image pixels unlimitedly, then the attack results would be even more damaging. For instance, in the NeRF-Synthetic (ship) scene, **the peak GPU usage could increase up to 21 times and almost reach 80 GB GPU memory**. This level approaches the upper limits of the most advanced GPUs currently available, such as H100 and A100, and is undeniably destructive.

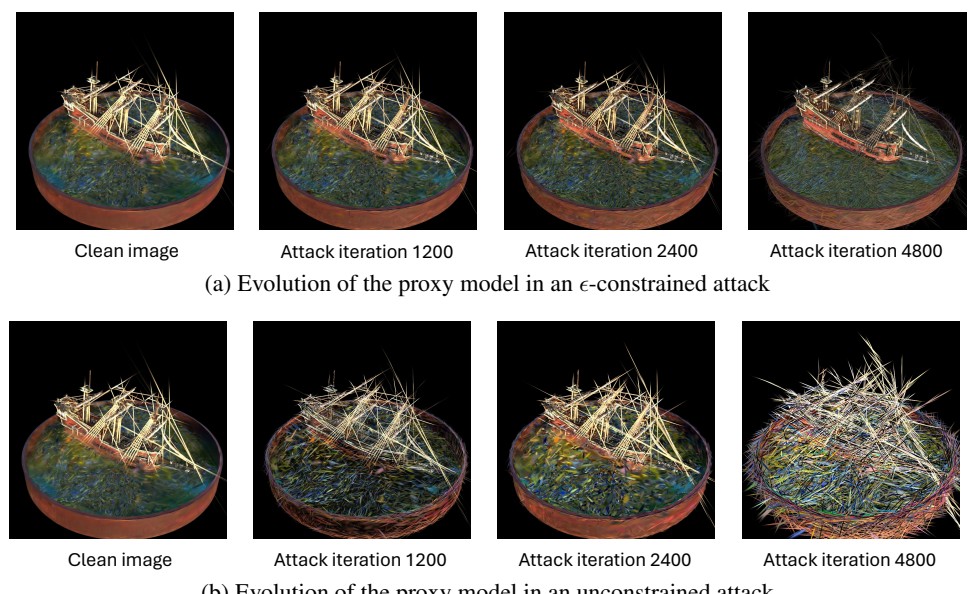

(a) Evolution of the proxy model in an $\epsilon$-constrained attack

(b) Evolution of the proxy model in an unconstrained attack

Figure 4: Visualizations of proxy model updates during an attack process. The proxy 3DGS model gradually obtains more complexity from learning from non-smoothness in the 2D image space.

To help the readers build more understanding into the attack procedure, we visualize the internal states during the attack optimization procedure, rendering the proxy model in Figure 4. It is evident that the proxy model can be guided from non-smoothness of 2D images to develop highly complex

Table 1: Attack results of *Poison-splat* across multiple datasets. Constrained attacks ($\epsilon = 16/255$) can covertly increase training costs through minor perturbations to images. In contrast, unconstrained attacks allow for unlimited modifications to the original input, leading to significantly higher resource consumption and more extensive damage. We highlight attacks which successfully consume over **24 GB GPU memory**, which can cause a denial-of-service(*i.e.*, out-of-memory error and service halting) if 3DGS service is provided on a common 24GB-memory GPU (*e.g.*, RTX 3090, RTX 4090 or A5000). Full results on all scenes provided in Table 3 and Table 4.

| | Number of Gaussians | | Peak GPU memory [MB] | | Training time [minutes] | | Render speed [FPS] | |
|---|---|---|---|---|---|---|---|---|
| **Constrained Poison-splat attack with $\epsilon = 16/255$** | | | | | | | | |
| Metric | | | | | | | | |
| Setting / Scene | clean | poisoned | clean | poisoned | clean | poisoned | clean | poisoned |
| NS-hotdog | 0.185 M | 1.147 M (6.20x↑) | 3336 | **29747**\* (8.92x↑) | 8.57 | 17.43 (2.03x↑) | 443 | 69 (6.42x↓) |
| NS-lego | 0.341 M | 0.805 M (2.36x↑) | 3532 | 9960 (2.82x↑) | 8.62 | 13.02 (1.51x↑) | 349 | 145 (2.41x↓) |
| NS-materials | 0.169 M | 0.410 M (2.43x↑) | 3172 | 4886 (1.54x↑) | 7.33 | 9.92 (1.35x↑) | 447 | 250 (1.79x↓) |
| NS-ship | 0.272 M | 1.071 M (3.94x↑) | 3692 | 16666 (4.51x↑) | 8.87 | 16.12 (1.82x↑) | 326 | 87 (3.75x↓) |
| TT-Courthouse | 0.604 M | 0.733 M (1.21x↑) | 11402 | 12688 (1.11x↑) | 13.81 | 14.49 (1.05x↑) | 284 | 220 (1.29x↓) |
| TT-Courtroom | 2.890 M | 5.450 M (1.89x↑) | 9896 | 16308 (1.65x↑) | 16.88 | 23.92 (1.42x↑) | 139 | 83 (1.67x↓) |
| TT-Museum | 4.439 M | 7.130 M (1.61x↑) | 12704 | 18790 (1.48x↑) | 19.79 | 26.38 (1.33x↑) | 121 | 78 (1.55x↓) |
| TT-Playground | 2.309 M | 4.307 M (1.87x↑) | 8717 | 13032 (1.50x↑) | 15.17 | 21.82 (1.44x↑) | 151 | 87 (1.74x↓) |
| MIP-bicycle | 5.793 M | 10.129 M (1.75x↑) | 17748 | **27074**\* (1.53x↑) | 33.42 | 44.44 (1.33x↑) | 69 | 43 (1.60x↓) |
| MIP-counter | 1.195 M | 4.628 M (3.87x↑) | 10750 | **29796**\* (2.77x↑) | 26.46 | 37.63 (1.42x↑) | 164 | 51 (3.22x↓) |
| MIP-room | 1.513 M | 7.129 M (4.71x↑) | 12316 | **46238**\* (3.75x↑) | 25.36 | 47.98 (1.89x↑) | 154 | 35 (4.40x↓) |
| MIP-stump | 4.671 M | 10.003 M (2.14x↑) | 14135 | **25714**\* (1.82x↑) | 27.06 | 38.48 (1.42x↑) | 111 | 57 (1.95x↓) |
| **Unconstrained Poison-splat attack with $\epsilon = \infty$** | | | | | | | | |
| Metric | | | | | | | | |
| Setting / Scene | clean | poisoned | clean | poisoned | clean | poisoned | clean | poisoned |
| NS-hotdog | 0.185 M | 4.272 M (23.09x↑) | 3336 | **47859**\* (14.35x↑) | 8.57 | 38.85 (4.53x↑) | 443 | 29 (15.28x↓) |
| NS-lego | 0.341 M | 4.159 M (12.20x↑) | 3532 | **78852**\* (22.33x↑) | 8.62 | 42.46 (4.93x↑) | 349 | 25 (13.96x↓) |
| NS-mic | 0.205 M | 3.940 M (19.22x↑) | 3499 | **61835**\* (17.67x↑) | 8.08 | 39.02 (4.83x↑) | 300 | 29 (10.34x↓) |
| NS-ship | 0.272 M | 4.317 M (15.87x↑) | 3692 | **80956**\* (21.93x↑) | 8.87 | 44.11 (4.97x↑) | 326 | 24 (13.58x↓) |
| TT-Courthouse | 0.604 M | 3.388 M (5.61x↑) | 11402 | **29856**\* (2.62x↑) | 13.81 | 25.33 (1.83x↑) | 284 | 54 (5.26x↓) |
| TT-Courtroom | 2.890 M | 13.196 M (4.57x↑) | 9896 | **33871**\* (3.42x↑) | 16.88 | 41.69 (2.47x↑) | 139 | 41 (3.39x↓) |
| TT-Museum | 4.439 M | 16.501 M (3.72x↑) | 12704 | **43317**\* (3.41x↑) | 19.79 | 48.89 (2.47x↑) | 121 | 36 (3.36x↓) |
| TT-Playground | 2.309 M | 10.306 M (4.46x↑) | 8717 | **27304**\* (3.13x↑) | 15.17 | 38.77 (2.56x↑) | 151 | 39 (3.87x↓) |
| MIP-bicycle | 5.793 M | 25.268 M (4.36x↑) | 17748 | **63236**\* (3.56x↑) | 33.42 | 81.48 (2.44x↑) | 69 | 16 (4.31x↓) |
| MIP-counter | 1.195 M | 11.167 M (9.34x↑) | 10750 | **80732**\* (7.51x↑) | 26.46 | 62.04 (2.34x↑) | 164 | 19 (8.63x↓) |
| MIP-room | 1.513 M | 16.019 M (10.59x↑) | 12316 | **57540**\* (4.67x↑) | 25.36 | 76.25 (3.01x↑) | 154 | 17 (9.06x↓) |
| MIP-stump | 4.671 M | 13.550 M (2.90x↑) | 14135 | **36181**\* (2.56x↑) | 27.06 | 51.51 (1.90x↑) | 111 | 27 (4.11x↓) |

3D shapes. Consequently, the poisoned data produced from the projection of this over-densified proxy model can produce more poisoned data, inducing more Gaussians to fit these poisoned data.

## 4.2 BLACK-BOX ATTACK PERFORMANCE

The attack results described previously assume a white-box scenario, where the attacker is fully aware of the victim system's details. In real-world, companies may utilize proprietary algorithms of their own, where the attacker may have no knowledge about their hyper-parameter settings or even their underlying 3DGS representations. This raises an important question: Can *Poison-splat* remain effective against black-box victim systems?

The answer is affirmative. To test the effectiveness of our attack on a black-box victim, we chose Scaffold-GS (Lu et al., 2024b), a 3DGS variant algorithm designed with a focus on efficiency. Scaffold-GS differs significantly from traditional Gaussian Splatting as it utilizes anchor points to distribute local 3D Gaussians, which greatly alters the underlying feature representation. We directly tested Scaffold-GS on the poisoned dataset we crafted for the traditional 3DGS algorithm, making the victim (Scaffold-GS trainer) a black-box model. The results, as presented in Table 2 and Table 5, demonstrate the attack's capability to generalize effectively, as evidenced by increased Gaussian parameters, higher GPU memory consumption, and reduced training speed.

The key reason for the success of generalization ability to black-box victims is our core intuition: promoting the non-smoothness of input images while maintaining their 3D consistency. This approach

Table 2: Black-box attack results on NeRF-Synthetic and MIP-NeRF360 dataset. The victim system, Scaffold-GS, utilizes distinctly different Gaussian Splatting feature representations compared to traditional Gaussian Splatting and remains unknown to the attacker. These results demonstrate the robust generalization ability of our attack against unknown black-box victim systems. Full results provided in Table 5.

| Metric | Number of Gaussians | | Peak GPU memory [MB] | | Training time [minutes] | |
|---|---|---|---|---|---|---|
| Scene     Setting | clean | poisoned | clean | poisoned | clean | poisoned |
| Constrained Black-box Poison-splat attack with $\epsilon = 16/255$ against Scaffold-GS | | | | | | |
| NS-lego | 0.414 M | 2.074 M (5.01x↑) | 3003 | 4808 (1.60x↑) | 9.77 | 17.91 (1.83x↑) |
| NS-ship | 1.000 M | 3.291 M (3.29x↑) | 3492 | 5024 (1.44x↑) | 11.68 | 21.92 (1.88x↑) |
| MIP-bonsai | 4.368 M | 10.608 M (2.43x↑) | 10080 | 12218 (1.21x↑) | 35.33 | 37.71 (1.07x↑) |
| MIP-stump | 6.798 M | 14.544 M (2.14x↑) | 7322 | 9432 (1.29x↑) | 33.53 | 36.32 (1.08x↑) |
| Unconstrained Black-box Poison-splat attack with $\epsilon = \infty$ against Scaffold-GS | | | | | | |
| NS-lego | 0.414 M | 3.973 M (9.60x↑) | 3003 | 6242 (2.08x↑) | 9.77 | 26.11 (2.67x↑) |
| NS-ship | 1.000 M | 4.717 M (4.72x↑) | 3492 | 6802 (1.95x↑) | 11.68 | 28.22 (2.42x↑) |
| MIP-bonsai | 4.368 M | 28.042 M (6.42x↑) | 10080 | 22115 (2.19x↑) | 35.33 | 78.36 (2.22x↑) |
| MIP-stump | 6.798 M | 34.027 M (5.01x↑) | 7322 | 20797 (2.84x↑) | 33.53 | 79.64 (2.38x↑) |

ensures that our poisoned dataset effectively increases the number of 3D-representation units for all adaptive complexity 3D reconstruction algorithms, regardless of the specific algorithm used.

## 4.3 ABLATION STUDY ON PROXY MODEL

We next ablate the effectiveness of our proxy model. As shown in Figure 3(b), we compare the number of Gaussians produced in victim learning procedures with two baselines: the green curve represents trajectory training on clean data, and the yellow curve represents a naive Total Variation maximizing attack, which does not involve a proxy model, but only maximize Total Variation score on each view individually. As we show in Figure 3(b), this naive attack is not as effective as our attack design. We attribute this reason to be lack of consistency of multi-view information. Since the Total Variation loss is a local image operator, individually maximizing the TV loss on each image will result in a noised dataset without aligning with each other. As a result, victim learning from this noised set of images struggles to align shapes in 3D space, and the training on fine details becomes harder to converge. Thus, when faced with conflicting textures from different views, the Gaussian Splatting technique fails to effectively perform reconstruction on details, leading to fewer Gaussians produced compared with our attack. We use this study to highlight the necessity of decoy model in our attack methodology design.

## 4.4 DISCUSSIONS ON NAIVE DEFENSE STRATEGIES

Our attack method maximizes the computational costs of 3DGS by optimizing the number of Gaussians. One may argue that a straightforward defense method is to simply impose an upper limit on the number of Gaussians during victim training to thwart the attack. However, such a straightforward strategy is not as effective as expected. While limiting the number of Gaussians might mitigate the direct impact on computational costs, it significantly degrades the quality of 3D reconstruction, as demonstrated in Figure 5. The figure shows that 3DGS models trained under these defensive constraints perform much worse compared to those with unconstrained training, particularly in terms of detail reconstruction. This decline in quality occurs because **3DGS cannot automatically distinguish necessary fine details from poisoned textures**. Naively capping the number of Gaussians will directly lead to the failure of the model to reconstruct the 3D scene accurately, which violates the primary goal of the service provider. This study demonstrates more sophisticated defensive strategies are necessary to both protect the system and maintain the quality of 3D reconstructions under our attack.

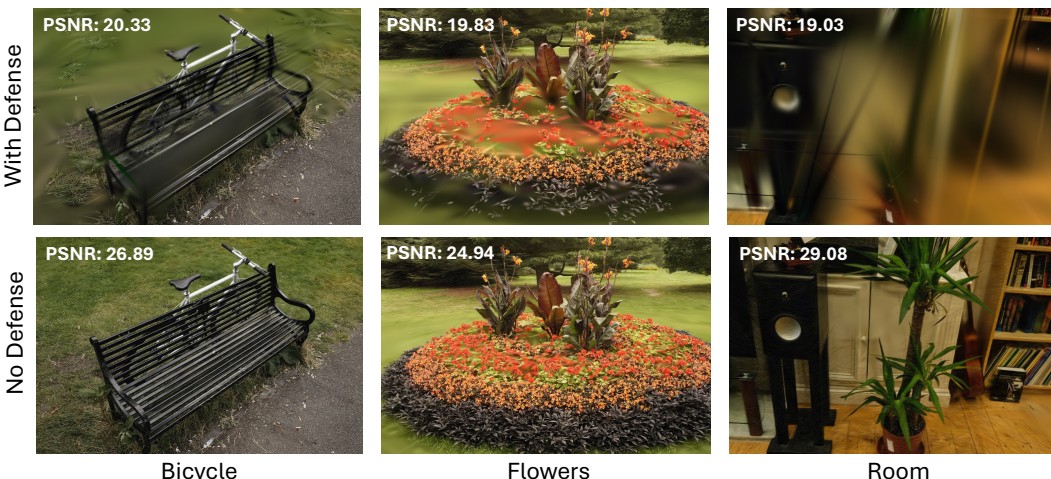

Figure 5: *Poison-splat* attack cannot be painlessly defended by simply constraining the number of Gaussians during 3DGS optimization. As demonstrated, while this defense can cap the maximum resource consumption, it markedly degrades the quality of 3D reconstruction, in terms of photo-realism and fine-grained details of the rendered images, which violates the primary goal of the service provider.

## 5 CONCLUSION AND DISCUSSIONS

This paper uncovers a significant yet previously overlooked security vulnerability in 3D Gaussian Splatting (3DGS). To this end, we have proposed a novel *Poison-splat* attack method, which dramatically escalates the computational demands of 3DGS and can even trigger a Denial-of-Service (*e.g.*, server disruption), causing substantial financial losses for 3DGS service providers. By leveraging a sophisticated bi-level optimization framework and strategies such as attack objective approximation, proxy model rendering, and optional constrained optimization, we demonstrate the feasibility of such attacks and emphasize the challenges in countering them with naive defenses. To the best of our knowledge, this is the first research work to explore an attack on 3DGS, and the first attack targeting the computation complexity of training phase in machine learning systems. We encourage researchers and practitioners in the field of 3DGS to acknowledge this security vulnerability and work together towards more robust algorithms and defense strategies against such threats. Lastly, we would like to discuss potential limitations, future directions, and social impacts of our work.

**Limitations and Future directions**. There are still limitations of our work, and we expect future studies can work on them:

1. **Better approximation of outer maximum optimization**. In this work, we approximate the outer maximum objective (*i.e.*, the computational cost) with the number of Gaussians. Although the number of Gaussians strongly correlates with GPU memory occupancy and rendering latency, there could be better optimizable metrics for this purpose. For example, the "density" of Gaussians, *i.e.*, the number of Gaussians falling into the same tile to involve in alpha-blending, might be a better metric to achieve better optimization outcomes.

2. **Better defense methods**. We focus on developing attack methods, without delving deeply into defensive strategies. We hope future studies can propose more robust 3DGS algorithms or develop more effective defensive techniques to counteract such attacks. This focus could significantly enhance the security and reliability of 3DGS systems in practical applications.

**Ethics Statement**. While our methods could potentially be misused by malicious actors to disrupt 3DGS service providers and cause financial harm, our aim is not to facilitate such actions. Instead, our objective is to highlight significant security vulnerabilities within 3DGS systems and prompt the community—researchers, practitioners, and providers—to recognize and address these gaps. We aim to inspire the development of more robust algorithms and defensive strategies, enhancing the security and reliability of 3DGS systems in practical applications. We are committed to ethical research and do not endorse using our findings to inflict societal harm.

## 6 ACKNOWLEDGEMENT

This project is supported in part by NUS Start-up Grant A-0010106-00-00, and by the National Research Foundation, Singapore, and Cyber Security Agency of Singapore under its National Cybersecurity R&D Programme and CyberSG R&D Cyber Research Programme Office (Award: CRPO-GC1-NTU-002).

We sincerely thank Xuanyu Yi (NTU), Xinjie Zhang (HKUST), Jiaqi Ma (WHU) and Shizun Wang (NUS) for their helpful discussions.

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

APPENDIX

The Appendix is organized as follows: Section A discusses related works; Section B provides a formal and comprehensive description of 3D Gaussian Splatting training. Section C explicitly describes the threat model in this paper, aiming to provide readers with a clearer understanding. Section D offers the full results of experiments mentioned in the main paper, presented here due to page constraints. Section E lists all the notations used throughout the paper. Section F presents examples of attacker-crafted poisoned images and the corresponding reconstructions from victim models trained with those images. Section G reports the attack performance under various levels of perturbation intensity, ranging from $\epsilon = 8/255$ to $\epsilon = 24/255$. Section H studies how effective the Poison-Splat attack will be if the poison ratio is not 100%. Section I shows the defensive threshold of limiting Gaussian numbers and discuss the difficulty of setting a uniform defense threshold. Section J discusses another potential defense: image smoothing as a pre-processing, and shows that it is not an ideal defense in terms of keeping reconstruction quality. Section K reports and analyzes the time complexity of Poison-Splat attack.

## A  RELATED WORKS

### A.1  3D GAUSSIAN SPLATTING

3D Gaussian Splatting is receiving overwhelming popularity in the 3D vision domain (Chen & Wang, 2024; Fei et al., 2024; Wu et al., 2024b; Dalal et al., 2024). The explicit nature of 3D Gaussian Splatting enables real-time rendering capabilities and unparalleled levels of control and editability, demonstrating its effectiveness in 3D generation. Since its introduction in (Kerbl et al., 2023), 3D Gaussian Splatting has facilitated a diverse array of downstream applications, including text-to-3D generation (Chen et al., 2024; Tang et al., 2024; Yi et al., 2024), Simultaneous Localization and Mapping (Keetha et al., 2024; Matsuki et al., 2024; Yan et al., 2024a), human avatars (Kocabas et al., 2024; Lei et al., 2024; Zielonka et al., 2023), 4D-dynamic scene modeling (Luiten et al., 2024; Wu et al., 2024a; Yang et al., 2024), and other applications (Xie et al., 2024b; Shen et al., 2024; Ye et al., 2023; Wang et al., 2024b;a).

While the majority of the audience in the 3D vision community places their focus on utility, we found only a handful of research works put efforts into investigating the security issues of 3D algorithms. Xie et al. (2024a) analyzed the resilience of 3D object detection systems under white-box and black-box attacks. (Fu et al., 2023; Horváth & Józsa, 2023) proposed attacks against Neural Radiance Fields (NeRFs). To our knowledge, this is the first attack ever proposed for 3D Gaussian Splatting systems.

### A.2  ATTACKS ON MACHINE LEARNING SYSTEMS

Machine learning has been shown to be vulnerable to various types of attacks (Pitropakis et al., 2019; Rigaki & Garcia, 2023; Chakraborty et al., 2018; Oliynyk et al., 2023; Tian et al., 2022; Zhang et al., 2021; 2023). The majority of attacks target the either *confidentiality* (including membership inference (Shokri et al., 2017; Salem et al., 2019; Choquette-Choo et al., 2021), data reconstruction attacks (Fredrikson et al., 2015; Carlini et al., 2021; 2023; Yu et al., 2023; Geiping et al., 2020; Lu et al., 2022) and model stealing attacks (Tramèr et al., 2016; Chandrasekaran et al., 2020)) or *integrity* (like adversarial attacks (Biggio et al., 2013; Szegedy et al., 2013; Dong et al., 2018; 2019; Lu et al., 2024a; Zhou et al.) and data poisoning attacks (Barreno et al., 2010; Jagielski et al., 2018; Biggio et al., 2012; Mei & Zhu, 2015b)). Different from the above mentioned attacks, we aim to illustrate that by carefully crafting input images, the attacker can indeed attack *availability*, *i.e.*, timely and cost-affordable access to machine learning service. The most similar research is the work by Shumailov et al. (2021), which is the first Denial-of-Service (DoS) attack towards machine learning systems. It proposed sponge examples which can cause much more latency and energy consumption for language models and image classifiers, driving deep learning models to their worst performance. Different from the work (Shumailov et al., 2021), our attack targets 3D reconstruction algorithms and is not driven by neural networks.

### A.2.1 DATA POISONING ATTACKS

We consider our proposed attacks to fall into the category of data poisoning attack, which manipulates training data to launch the attack at the training stage. The attacker makes the first move to craft poisoned training samples, and the victim has a goal of training the system based on samples provided by the attacker. Solving this problem typically involves solving a max-min bi-level optimization problem, where the attacker wants to maximize some adversarial loss (*i.e.*, test error or computational cost) and victim wants to minimize the training objective (*i.e.*, training loss). Different literature solves the problem using different techniques (Biggio et al., 2012; Mei & Zhu, 2015b;a; Li et al., 2016). For many machine learning systems, given the inherent complexity and their reliance on large datasets, it is intractable to exactly do the bi-level optimization. The work (Muñoz-González et al., 2017) exploited back-gradient optimization technique to approximate real gradients through reverse-mode differentiation. The work (Huang et al., 2020) formulated as a meta-learning problem, and approximated the inner optimal solution by training only $K$ SGD steps for each outer objective evaluation. The work (Koh et al., 2022) proposed three methods to approximate the expensive bi-level optimization: influence functions (Koh & Liang, 2017), minimax duality and Karush-Kuhn-Tucker (KKT) conditions. The work (Fowl et al., 2021a) used gradient matching which assumes inner optimality has been achieved by a fixed model. The work (Fowl et al., 2021b) avoided the bi-level optimization with a well-trained proxy model from clean data. Different from these previous studies, we propose a gradient-free optimization process to solve the outer optimization pipeline, and show a good optimization result on the attacker's goal.

### A.2.2 ENERGY-LATENCY ATTACKS IN MACHINE LEARNING

A limited number of studies have investigated attacks to manipulate the computation cost needed for machine learning systems, mainly in inference-stage. Shumailov et al. (2021) initially brought people's attention to sponge samples, which are malicious samples able to maximize the energy consumption and responsive latency of a neural network. Following this work, there are other works studying samples which can cause the worst case computation resource consumption in a (dynamic) machine learning systems, for example Hong et al. (2020) and Chen et al. (2023) found samples that can necessitate late exit in a dynamic-depth neural network. More up-to-date works utilize the flexible nature of language to attack NLP models, inducing extended natural language sequence responses to over-consume energy and latency (Chen et al., 2022b;a; Gao et al., 2023). However, to the best of our knowledge, there has been **no previous work targeting Denial-of-Service attacks at the training stage** of machine learning systems. Our work is pioneering in proposing a DoS attack specifically designed for the training phase of machine learning systems. Additionally, this is also the first effort to develop an attack against 3D Gaussian Splatting.

## B   MORE BACKGROUNDS OF 3D GAUSSIAN SPLATTING

3D Gaussian Splatting (3DGS) (Kerbl et al., 2023) has revolutionized the 3D vision community with its superior performance and efficiency. It models a 3D scene by learning a set of 3D Gaussians $\mathcal{G} = \{G_i\}$ from multi-view images $\mathcal{D} = \{V_k, P_k\}_{k=1}^N$, where each view consists of a ground-truth image $V_k$ and the corresponding camera poses $P_k$. We denote $N$ as the total number of image-pose pairs in the training data, which can be arbitrary and unfixed in 3DGS. As for the model, each 3D Gaussian $G_i = \{\mu_i, s_i, \alpha_i, q_i, c_i\}$ has 5 optimizable parameters: center coordinates $\mu_i \in \mathbb{R}^3$, scales $s_i \in \mathbb{R}^3$, opacity $\alpha_i \in \mathbb{R}$, rotation quaternion $q_i \in \mathbb{R}^4$, and associated view-dependent color represented as spherical harmonics $c_i \in \mathbb{R}^{3(d+1)^2}$, where $d$ is the degree of spherical harmonics.

To render $\mathcal{G}$ from 3D Gaussian space to 2D images, each Gaussian is first projected into the camera coordinate frame given a camera pose $P_k$ to determine the depth of each Gaussian, which is the distance of that Gaussian to the camera view. The calculated depth is used for sorting Gaussians from near to far, and then the colors in 2D image are rendered in parallel by alpha blending according to the depth order of adjacent 3D Gaussians. The core reason for 3DGS to achieve real-time optimization and rendering lies in this tile-based differentiable rasterization pipeline. We denote the 3DGS renderer as a function $\mathcal{R}$, which takes as input the Gaussians $\mathcal{G}$ and a camera pose $P_k$ to render an image $V_k' = \mathcal{R}(\mathcal{G}, P_k)$.

The training is done by optimizing the reconstruction loss between the rendered image and ground truth image for each camera view, where the reconstruction loss is a combination of $\mathcal{L}_1$ loss and structural similarity index measure (SSIM) loss $\mathcal{L}_{\text{D-SSIM}}$:

$$\min_{\mathcal{G}} \mathcal{L}(\mathcal{D}) = \min_{\mathcal{G}} \sum_{k=1}^{N} \mathcal{L}(V_k', V_k) = \min_{\mathcal{G}} \sum_{k=1}^{N} (1-\lambda)\mathcal{L}_1(V_k', V_k) + \lambda \mathcal{L}_{\text{D-SSIM}}(V_k', V_k), \quad (6)$$

where $\lambda$ is a trade-off parameter. To ensure finer reconstruction, Gaussian Splatting deploys *adaptive density control* to automatically add or remove Gaussians from optimization. The process of adding Gaussians, known as densification, involves two primary operations: splitting and cloning. Conversely, removing Gaussians, called pruning, indicates eliminating a Gaussian from the set of optimization variables. Both densification and pruning are non-differentiable operations. The adaptive density control is performed during regular optimization intervals and follows specific, predefined rules. In the initial design (Kerbl et al., 2023), for a Gaussian $G_i$ with view-space projection point $\mu_i^k = (\mu_{i,x}^k, \mu_{i,y}^k)$ under viewpoint $V_k$, the average view-space positional gradient $\nabla_{\mu_i} \mathcal{L}$ is calculated every 100 training iterations (Ye et al., 2024):

$$\nabla_{\mu_i} \mathcal{L} = \frac{\sum_{k=1}^{M} \left\| \frac{\partial \mathcal{L}(V_k', V_k)}{\partial \mu_i^k} \right\|}{M}, \quad (7)$$

where $M$ is the total number of views that the Gaussian $G_i$ participates in calculation during the 100 iterations (Ye et al., 2024) and $M$ is usually less than $N$. Densification is performed when the view-space positional gradient $\nabla_{\mu_i} \mathcal{L}$ is larger than a pre-set gradient threshold $\tau_g$ (by default 0.0002). Moreover, the decision to split or clone a Gaussian $G_i$ is based on its scale vector $s_i$, which is compared with a scale threshold $\tau_s$: it is split if larger, and cloned if smaller. Pruning occurs when the opacity $\alpha_i$ falls below the opacity threshold $\tau_\alpha$. We also notice some recent 3DGS studies (Cheng et al., 2024; Fang & Wang, 2024; Li et al., 2024; Bulò et al., 2024) that adopt different yet still pre-defined densification rules.

## C    THREAT MODEL

We provide a structured description of the attack's goal, input, output, information and constraints in this section, to provide more clarity of the threat model:

**Attacker Goal**: To increase the computation cost and over-consume computation resource of 3DGS service provider.

**Attacker Input**: Clean data.

**Attacker Output**: Poisoned data by running attack on the clean data.

**Attacker Information**: Attacker doesn't need to know the underlying hardware device of the victim.

- White-box: Attacker knows the specific 3DGS configuration of the victim.
- Black-box: Attacker only knows victim is using 3DGS, but does not know which variant of 3DGS representation and what configuration the victim is using.

**Attacker Constraint**: Attacker can optionally choose to constrain the perturbation range $\epsilon$ of his additive perturbation to add on clean data.

**Attacker Algorithm**: Algorithm 1.

Table 3: Full constrained attack results ($\epsilon = 16/255$) on all scenes from the NeRF-Synthetic, MIP-NeRF360 and Tanks-and-Temples datasets. All results are reported by averaging over three individual runs.

| | \multicolumn{8}{c|}{Constrained Poison-splat attack with $\epsilon = 16/255$} |
|---|---|---|---|---|---|---|---|---|
| Metric | \multicolumn{2}{c|}{Number of Gaussians} | \multicolumn{2}{c|}{Peak GPU memory [MB]} | \multicolumn{2}{c|}{Training time [minutes]} | \multicolumn{2}{c}{Render speed [FPS]} |
| Setting / Scene | clean | poisoned | clean | poisoned | clean | poisoned | clean | poisoned |
| NS-chair | 0.494 M | 0.957 M (1.94x↑) | 3752 | 8003 (2.13x↑) | 9.45 | 13.41 (1.42x↑) | 260 | 117 (2.22x↓) |
| NS-drums | 0.385 M | 0.695 M (1.81x↑) | 3456 | 7190 (2.08x↑) | 8.75 | 11.94 (1.36x↑) | 347 | 144 (2.41x↓) |
| NS-ficus | 0.265 M | 0.273 M (1.03x↑) | 3194 | 3783 (1.18x↑) | 6.99 | 8.94 (1.28x↑) | 346 | 337 (1.03x↓) |
| NS-hotdog | 0.185 M | 1.147 M (6.20x↑) | 3336 | 29747* (8.92x↑) | 8.57 | 17.43 (2.03x↑) | 443 | 69 (6.42x↓) |
| NS-lego | 0.341 M | 0.805 M (2.36x↑) | 3532 | 9960 (2.82x↑) | 8.62 | 13.02 (1.51x↑) | 349 | 145 (2.41x↓) |
| NS-materials | 0.169 M | 0.410 M (2.43x↑) | 3172 | 4886 (1.54x↑) | 7.33 | 9.92 (1.35x↑) | 447 | 250 (1.79x↓) |
| NS-mic | 0.205 M | 0.359 M (1.75x↑) | 3499 | 6437 (1.84x↑) | 8.08 | 10.96 (1.36x↑) | 300 | 172 (1.74x↓) |
| NS-ship | 0.272 M | 1.071 M (3.94x↑) | 3692 | 16666 (4.51x↑) | 8.87 | 16.12 (1.82x↑) | 326 | 87 (3.75x↓) |
| MIP-bicycle | 5.793 M | 10.129 M (1.75x↑) | 17748 | 27074* (1.53x↑) | 33.42 | 44.44 (1.33x↑) | 69 | 43 (1.60x↓) |
| MIP-bonsai | 1.294 M | 6.150 M (4.75x↑) | 10216 | 20564 (2.01x↑) | 21.68 | 32.51 (1.50x↑) | 214 | 66 (3.24x↓) |
| MIP-counter | 1.195 M | 4.628 M (3.87x↑) | 10750 | 29796* (2.77x↑) | 26.46 | 37.63 (1.42x↑) | 164 | 51 (3.22x↓) |
| MIP-flowers | 3.508 M | 5.177 M (1.48x↑) | 12520 | 16338 (1.30x↑) | 25.47 | 30.05 (1.18x↑) | 124 | 89 (1.39x↓) |
| MIP-garden | 5.684 M | 7.697 M (1.35x↑) | 17561 | 23115 (1.32x↑) | 33.19 | 40.82 (1.23x↑) | 81 | 54 (1.50x↓) |
| MIP-kitchen | 1.799 M | 6.346 M (3.53x↑) | 12442 | 20902 (1.68x↑) | 26.61 | 43.24 (1.62x↑) | 133 | 46 (2.89x↓) |
| MIP-room | 1.513 M | 7.129 M (4.71x↑) | 12316 | 46238* (3.75x↑) | 25.36 | 47.98 (1.89x↑) | 154 | 35 (4.40x↓) |
| MIP-stump | 4.671 M | 10.003 M (2.14x↑) | 14135 | 25714* (1.82x↑) | 27.06 | 38.48 (1.42x↑) | 111 | 57 (1.95x↓) |
| MIP-treehill | 3.398 M | 5.684 M (1.67x↑) | 11444 | 17287 (1.51x↑) | 22.76 | 29.26 (1.29x↑) | 123 | 70 (1.76x↓) |
| TT-Auditorium | 0.699 M | 2.719 M (3.89x↑) | 5982 | 14532 (2.43x↑) | 12.69 | 19.05 (1.50x↑) | 266 | 100 (2.66x↓) |
| TT-Ballroom | 3.103 M | 3.957 M (1.28x↑) | 10416 | 12226 (1.17x↑) | 20.37 | 22.43 (1.10x↑) | 96 | 85 (1.13x↓) |
| TT-Barn | 1.005 M | 1.865 M (1.86x↑) | 7813 | 10470 (1.34x↑) | 13.10 | 16.27 (1.24x↑) | 229 | 127 (1.80x↓) |
| TT-Caterpillar | 1.290 M | 1.857 M (1.44x↑) | 6624 | 8014 (1.21x↑) | 11.96 | 14.77 (1.23x↑) | 228 | 164 (1.39x↓) |
| TT-Church | 2.351 M | 3.396 M (1.44x↑) | 10076 | 11797 (1.17x↑) | 17.43 | 20.54 (1.18x↑) | 132 | 97 (1.36x↓) |
| TT-Courthouse | 0.604 M | 0.733 M (1.21x↑) | 11402 | 12688 (1.11x↑) | 13.81 | 14.49 (1.05x↑) | 284 | 220 (1.29x↓) |
| TT-Courtroom | 2.890 M | 5.450 M (1.89x↑) | 9896 | 16308 (1.65x↑) | 16.88 | 23.92 (1.42x↑) | 139 | 83 (1.67x↓) |
| TT-Family | 2.145 M | 3.734 M (1.74x↑) | 7261 | 10707 (1.47x↑) | 13.58 | 18.46 (1.36x↑) | 182 | 111 (1.64x↓) |
| TT-Francis | 0.759 M | 1.612 M (2.12x↑) | 5208 | 6938 (1.33x↑) | 10.19 | 13.92 (1.37x↑) | 279 | 168 (1.66x↓) |
| TT-Horse | 1.308 M | 2.510 M (1.92x↑) | 5125 | 7762 (1.51x↑) | 12.05 | 16.47 (1.37x↑) | 220 | 121 (1.82x↓) |
| TT-Ignatius | 3.144 M | 3.937 M (1.25x↑) | 9943 | 11800 (1.19x↑) | 16.32 | 18.52 (1.13x↑) | 149 | 118 (1.26x↓) |
| TT-Lighthouse | 0.888 M | 1.178 M (1.33x↑) | 7466 | 8696 (1.16x↑) | 14.21 | 15.92 (1.12x↑) | 213 | 163 (1.31x↓) |
| TT-M60 | 1.667 M | 2.980 M (1.79x↑) | 7948 | 10874 (1.37x↑) | 14.08 | 18.22 (1.29x↑) | 200 | 107 (1.87x↓) |
| TT-Meetingroom | 1.267 M | 2.927 M (2.31x↑) | 6645 | 10234 (1.54x↑) | 13.53 | 18.97 (1.40x↑) | 197 | 102 (1.93x↓) |
| TT-Museum | 4.439 M | 7.130 M (1.61x↑) | 12704 | 18790 (1.48x↑) | 19.79 | 26.38 (1.33x↑) | 121 | 78 (1.55x↓) |
| TT-Palace | 0.711 M | 0.696 M (0.98x↑) | 8980 | 8372 (0.93x↑) | 14.27 | 14.54 (1.02x↑) | 185 | 186 (0.99x↓) |
| TT-Panther | 1.812 M | 3.457 M (1.91x↑) | 8511 | 12048 (1.42x↑) | 13.49 | 19.46 (1.44x↑) | 199 | 103 (1.93x↓) |
| TT-Playground | 2.309 M | 4.307 M (1.87x↑) | 8717 | 13032 (1.50x↑) | 15.17 | 21.82 (1.44x↑) | 151 | 87 (1.74x↓) |
| TT-Temple | 0.893 M | 1.345 M (1.51x↑) | 6929 | 7525 (1.09x↑) | 13.03 | 15.26 (1.17x↑) | 216 | 149 (1.45x↓) |
| TT-Train | 1.113 M | 1.332 M (1.20x↑) | 6551 | 7128 (1.09x↑) | 12.64 | 13.62 (1.08x↑) | 193 | 163 (1.18x↓) |
| TT-Truck | 2.533 M | 3.630 M (1.43x↑) | 8662 | 11014 (1.27x↑) | 14.64 | 20.12 (1.37x↑) | 156 | 88 (1.77x↓) |

# D EXTENDED EXPERIMENTS

## D.1 FULL ATTACK RESULTS

We report the constrained attack results with $\epsilon = 16/255$ on all scenes of NeRF-Synthetic, Mip-NeRF360 and Tanks-and-Temples dataset in Table 3, and the unconstrained attack results in Table 4.

Table 4: Full unconstrained attack results ($\epsilon = \infty$) on all scenes from the NeRF-Synthetic, MIP-NeRF360 and Tanks-and-Temples datasets. All results are reported by averaging over three individual runs.

| | | | | | | | | |
|---|---|---|---|---|---|---|---|---|
| **Unconstrained Poison-splat attack with $\epsilon = \infty$** | | | | | | | | |
| Metric | Number of Gaussians | | Peak GPU memory [MB] | | Training time [minutes] | | Render speed [FPS] | |
| Setting Scene | clean | poisoned | clean | poisoned | clean | poisoned | clean | poisoned |
| NS-chair | 0.494 M | 4.152 M (8.40x↑) | 3752 | **51424**\* (13.71x↑) | 9.45 | 40.71 (4.31x↑) | 260 | 27 (9.63x↓) |
| NS-drums | 0.385 M | 4.109 M (10.67x↑) | 3456 | **62328**\* (18.03x↑) | 8.75 | 43.25 (4.94x↑) | 347 | 24 (14.46x↓) |
| NS-ficus | 0.265 M | 3.718 M (14.03x↑) | 3194 | **50450**\* (15.80x↑) | 6.99 | 36.92 (5.28x↑) | 346 | 31 (11.16x↓) |
| NS-hotdog | 0.185 M | 4.272 M (23.09x↑) | 3336 | **47859**\* (14.35x↑) | 8.57 | 38.85 (4.53x↑) | 443 | 29 (15.28x↓) |
| NS-lego | 0.341 M | 4.159 M (12.20x↑) | 3532 | **78852**\* (22.33x↑) | 8.62 | 42.46 (4.93x↑) | 349 | 25 (13.96x↓) |
| NS-materials | 0.169 M | 3.380 M (20.00x↑) | 3172 | **43998**\* (13.87x↑) | 7.33 | 32.40 (4.42x↑) | 447 | 37 (12.08x↓) |
| NS-mic | 0.205 M | 3.940 M (19.22x↑) | 3499 | **61835**\* (17.67x↑) | 8.08 | 39.02 (4.83x↑) | 300 | 29 (10.34x↓) |
| NS-ship | 0.272 M | 4.317 M (15.87x↑) | 3692 | **80956**\* (21.93x↑) | 8.87 | 44.11 (4.97x↑) | 326 | 24 (13.58x↓) |
| MIP-bicycle | 5.793 M | 25.268 M (4.36x↑) | 17748 | **63236**\* (3.56x↑) | 33.42 | 81.48 (2.44x↑) | 69 | 16 (4.31x↓) |
| MIP-bonsai | 1.294 M | 20.127 M (15.55x↑) | 10216 | **54506**\* (5.34x↑) | 21.68 | 75.18 (3.47x↑) | 214 | 18 (11.89x↓) |
| MIP-counter | 1.195 M | 11.167 M (9.34x↑) | 10750 | **80732**\* (7.51x↑) | 26.46 | 62.04 (2.34x↑) | 164 | 19 (8.63x↓) |
| MIP-flowers | 3.508 M | 18.075 M (5.15x↑) | 12520 | **45515**\* (3.64x↑) | 25.47 | 62.62 (2.46x↑) | 124 | 24 (5.17x↓) |
| MIP-garden | 5.684 M | 21.527 M (3.79x↑) | 17561 | **52140**\* (2.97x↑) | 33.19 | 83.81 (2.53x↑) | 81 | 17 (4.76x↓) |
| MIP-kitchen | 1.799 M | 12.830 M (7.13x↑) | 12442 | **77141**\* (6.20x↑) | 26.61 | 73.04 (2.74x↑) | 133 | 16 (8.31x↓) |
| MIP-room | 1.513 M | 16.019 M (10.59x↑) | 12316 | **57540**\* (4.67x↑) | 25.36 | 76.25 (3.01x↑) | 154 | 17 (9.06x↓) |
| MIP-stump | 4.671 M | 13.550 M (2.90x↑) | 14135 | **36181**\* (2.56x↑) | 27.06 | 51.51 (1.90x↑) | 111 | 27 (4.11x↓) |
| MIP-treehill | 3.398 M | 13.634 M (4.01x↑) | 11444 | **36299**\* (3.17x↑) | 22.76 | 52.83 (2.32x↑) | 123 | 24 (5.12x↓) |
| TT-Auditorium | 0.699 M | 4.153 M (5.94x↑) | 5982 | 12666 (2.12x↑) | 12.69 | 22.50 (1.77x↑) | 266 | 76 (3.50x↓) |
| TT-Ballroom | 3.103 M | 7.534 M (2.43x↑) | 10416 | 19832 (1.90x↑) | 20.37 | 33.63 (1.65x↑) | 96 | 46 (2.09x↓) |
| TT-Barn | 1.005 M | 5.456 M (5.43x↑) | 7813 | 15682 (2.01x↑) | 13.10 | 26.10 (1.99x↑) | 229 | 61 (3.75x↓) |
| TT-Caterpillar | 1.290 M | 6.980 M (5.41x↑) | 6624 | 18909 (2.85x↑) | 11.96 | 29.51 (2.47x↑) | 228 | 53 (4.30x↓) |
| TT-Church | 2.351 M | 7.564 M (3.22x↑) | 10076 | 21296 (2.11x↑) | 17.43 | 33.39 (1.92x↑) | 132 | 43 (3.07x↓) |
| TT-Courthouse | 0.604 M | 3.388 M (5.61x↑) | 11402 | **29856**\* (2.62x↑) | 13.81 | 25.33 (1.83x↑) | 284 | 54 (5.26x↓) |
| TT-Courtroom | 2.890 M | 13.196 M (4.57x↑) | 9896 | **33871**\* (3.42x↑) | 16.88 | 41.69 (2.47x↑) | 139 | 41 (3.39x↓) |
| TT-Family | 2.145 M | 11.700 M (5.45x↑) | 7261 | **27533**\* (3.79x↑) | 13.58 | 38.79 (2.86x↑) | 182 | 43 (4.23x↓) |
| TT-Francis | 0.759 M | 7.435 M (9.80x↑) | 5208 | 19777 (3.80x↑) | 10.19 | 31.01 (3.04x↑) | 279 | 51 (5.47x↓) |
| TT-Horse | 1.308 M | 9.358 M (7.15x↑) | 5125 | 22756 (4.44x↑) | 12.05 | 34.90 (2.90x↑) | 220 | 46 (4.78x↓) |
| TT-Ignatius | 3.144 M | 11.278 M (3.59x↑) | 9943 | **28895**\* (2.91x↑) | 16.32 | 39.61 (2.43x↑) | 149 | 42 (3.55x↓) |
| TT-Lighthouse | 0.888 M | 5.081 M (5.72x↑) | 7466 | 23842 (3.19x↑) | 14.21 | 30.67 (2.16x↑) | 213 | 47 (4.53x↓) |
| TT-M60 | 1.667 M | 7.076 M (4.24x↑) | 7948 | 21062 (2.65x↑) | 14.08 | 31.82 (2.26x↑) | 200 | 46 (4.35x↓) |
| TT-Meetingroom | 1.267 M | 7.066 M (5.58x↑) | 6645 | 19182 (2.89x↑) | 13.53 | 31.44 (2.32x↑) | 197 | 50 (3.94x↓) |
| TT-Museum | 4.439 M | 16.501 M (3.72x↑) | 12704 | **43317**\* (3.41x↑) | 19.79 | 48.89 (2.47x↑) | 121 | 36 (3.36x↓) |
| TT-Palace | 0.711 M | 2.764 M (3.89x↑) | 8980 | 14065 (1.57x↑) | 14.27 | 20.07 (1.41x↑) | 185 | 84 (2.20x↓) |
| TT-Panther | 1.812 M | 8.112 M (4.48x↑) | 8511 | 22638 (2.66x↑) | 13.49 | 33.19 (2.46x↑) | 199 | 49 (4.06x↓) |
| TT-Playground | 2.309 M | 10.306 M (4.46x↑) | 8717 | **27304**\* (3.13x↑) | 15.17 | 38.77 (2.56x↑) | 151 | 39 (3.87x↓) |
| TT-Temple | 0.893 M | 5.231 M (5.86x↑) | 6929 | 15238 (2.20x↑) | 13.03 | 27.96 (2.15x↑) | 216 | 52 (4.15x↓) |
| TT-Train | 1.113 M | 4.916 M (4.42x↑) | 6551 | 14840 (2.27x↑) | 12.64 | 25.34 (2.00x↑) | 193 | 59 (3.27x↓) |
| TT-Truck | 2.533 M | 8.004 M (3.16x↑) | 8662 | 21166 (2.44x↑) | 14.64 | 33.26 (2.27x↑) | 156 | 47 (3.32x↓) |

## D.2 FULL BLACK-BOX ATTACK RESULTS

We report the full results of attacking a black-box victim (Scaffold-GS) (Lu et al., 2024b) on NeRF-Synthetic and MIP-NeRF360 datasets in Table 5.

To further test the effectiveness of Poison-Splat, we conduct black-box attack experiments on Mip-Splatting (Yu et al., 2024), which received the best student paper award at CVPR 2024 and has been highly popular (with 178 citations as of November 20th, 2024). Mip-Splatting is an advanced variant of the original 3D Gaussian Splatting, incorporating a 3D smoothing filter and a 2D Mip filter. These enhancements help eliminate various artifacts and achieve alias-free renderings. Results are shown in Table 6.

We found that Mip-Splatting (Yu et al., 2024) consumes more GPU memory compared to the original 3D Gaussian Splatting, making it more prone to the worst attack consequence of running Out-of-Memory. As illustrated in Table 6, even when the attack perturbation is constrained to $\epsilon = 16/255$, the GPU memory consumption nearly reached the 80 GB capacity of Nvidia A800. When we apply an unconstrained attack, all scenes in the MIP-NeRF360 dataset will result in denial-of-service.

Table 5: Full black-box attack results on NeRF-Synthetic and MIP-NeRF360 datasets. The victim system, Scaffold-GS, utilizes distinctly different Gaussian Splatting feature representations compared to traditional Gaussian Splatting and remains unknown to the attacker. These results demonstrate the robust generalization ability of our attack against unknown black-box victim systems.

| Constrained Black-box Poison-splat attack with $\epsilon = 16/255$ against Scaffold-GS | | | | | | |
|---|---|---|---|---|---|---|
| Metric | Number of Gaussians | | Peak GPU memory [MB] | | Training time [minutes] | |
| Setting
Scene | clean | poisoned | clean | poisoned | clean | poisoned |
| NS-chair | 1.000 M | 1.615 M (1.61x↑) | 3418 | 4003 (1.17x↑) | 9.91 | 17.18 (1.73x↑) |
| NS-drums | 1.000 M | 1.771 M (1.77x↑) | 4022 | 4382 (1.09x↑) | 10.62 | 16.29 (1.53x↑) |
| NS-ficus | 1.000 M | 2.301 M (2.30x↑) | 3416 | 4234 (1.24x↑) | 9.64 | 18.54 (1.92x↑) |
| NS-hotdog | 1.000 M | 1.463 M (1.46x↑) | 3539 | 4516 (1.28x↑) | 10.74 | 18.62 (1.73x↑) |
| NS-lego | 0.414 M | 2.074 M (5.01x↑) | 3003 | 4808 (1.60x↑) | 9.77 | 17.91 (1.83x↑) |
| NS-materials | 1.000 M | 2.520 M (2.52x↑) | 4110 | 5190 (1.26x↑) | 10.87 | 17.75 (1.63x↑) |
| NS-mic | 1.000 M | 2.053 M (2.05x↑) | 3682 | 5054 (1.37x↑) | 10.36 | 18.53 (1.79x↑) |
| NS-ship | 1.000 M | 3.291 M (3.29x↑) | 3492 | 5024 (1.44x↑) | 11.68 | 21.92 (1.88x↑) |
| MIP-bicycle | 8.883 M | 16.947 M (1.91x↑) | 12817 | 13650 (1.06x↑) | 38.93 | 44.98 (1.16x↑) |
| MIP-bonsai | 4.368 M | 10.608 M (2.43x↑) | 10080 | 12218 (1.21x↑) | 35.33 | 37.71 (1.07x↑) |
| MIP-counter | 2.910 M | 6.387 M (2.19x↑) | 12580 | 17702 (1.41x↑) | 38.53 | 44.83 (1.16x↑) |
| MIP-flowers | 7.104 M | 11.614 M (1.63x↑) | 8241 | 9766 (1.19x↑) | 36.56 | 38.31 (1.05x↑) |
| MIP-garden | 7.630 M | 11.555 M (1.51x↑) | 9194 | 10966 (1.19x↑) | 38.57 | 42.79 (1.11x↑) |
| MIP-kitchen | 3.390 M | 6.698 M (1.98x↑) | 14037 | 16895 (1.20x↑) | 43.83 | 48.10 (1.10x↑) |
| MIP-room | 2.846 M | 10.527 M (3.70x↑) | 13323 | 13981 (1.05x↑) | 36.24 | 48.01 (1.32x↑) |
| MIP-stump | 6.798 M | 14.544 M (2.14x↑) | 7322 | 9432 (1.29x↑) | 33.53 | 36.32 (1.08x↑) |
| **Unconstrained Black-box Poison-splat attack with $\epsilon = \infty$ against Scaffold-GS** | | | | | | |
| Metric | Number of Gaussians | | Peak GPU memory [MB] | | Training time [minutes] | |
| Setting
Scene | clean | poisoned | clean | poisoned | clean | poisoned |
| NS-chair | 1.000 M | 3.082 M (3.08x↑) | 3418 | 5197 (1.52x↑) | 9.91 | 25.34 (2.56x↑) |
| NS-drums | 1.000 M | 5.026 M (5.03x↑) | 4022 | 7364 (1.83x↑) | 10.62 | 28.24 (2.66x↑) |
| NS-ficus | 1.000 M | 2.106 M (2.11x↑) | 3416 | 4446 (1.30x↑) | 9.64 | 21.65 (2.25x↑) |
| NS-hotdog | 1.000 M | 3.541 M (3.54x↑) | 3539 | 5492 (1.55x↑) | 10.74 | 24.93 (2.32x↑) |
| NS-lego | 0.414 M | 3.973 M (9.60x↑) | 3003 | 6242 (2.08x↑) | 9.77 | 26.11 (2.67x↑) |
| NS-materials | 1.000 M | 3.997 M (4.00x↑) | 4110 | 5997 (1.46x↑) | 10.87 | 25.48 (2.34x↑) |
| NS-mic | 1.000 M | 3.021 M (3.02x↑) | 3682 | 5490 (1.49x↑) | 10.36 | 23.71 (2.29x↑) |
| NS-ship | 1.000 M | 4.717 M (4.72x↑) | 3492 | 6802 (1.95x↑) | 11.68 | 28.22 (2.42x↑) |
| MIP-bicycle | 8.883 M | 33.284 M (3.75x↑) | 12817 | 22042 (1.72x↑) | 38.93 | 84.71 (2.18x↑) |
| MIP-bonsai | 4.368 M | 28.042 M (6.42x↑) | 10080 | 22115 (2.19x↑) | 35.33 | 78.36 (2.22x↑) |
| MIP-counter | 2.910 M | 12.928 M (4.44x↑) | 12580 | 16168 (1.29x↑) | 38.53 | 55.59 (1.44x↑) |
| MIP-flowers | 7.104 M | 27.610 M (3.89x↑) | 8241 | 18352 (2.23x↑) | 36.56 | 73.43 (2.01x↑) |
| MIP-garden | 7.630 M | 23.828 M (3.12x↑) | 9194 | 20400 (2.22x↑) | 38.57 | 85.45 (2.22x↑) |
| MIP-kitchen | 3.390 M | 14.404 M (4.25x↑) | 14037 | 17838 (1.27x↑) | 43.83 | 63.32 (1.44x↑) |
| MIP-room | 2.846 M | 21.060 M (7.40x↑) | 13323 | 21672 (1.63x↑) | 36.24 | 76.94 (2.12x↑) |
| MIP-stump | 6.798 M | 34.027 M (5.01x↑) | 7322 | 20797 (2.84x↑) | 33.53 | 79.64 (2.38x↑) |

We examined the code implementation of Mip-Splatting[7], and identified that it uses quantile computation in its Gaussian model densification function [8] which requires massive GPU memory and easily triggers out-of-memory. Our attack highlights the vulnerability in various 3DGS algorithm implementations.

---

[7] https://github.com/autonomousvision/mip-splatting
[8] In Line 524 of `mip-splatting/scene/gaussian_model.py`

Table 6: Black-box attack results on Mip-Splatting (Yu et al., 2024) as the victim system, which can further demonstrate the robust generalization ability of our attack against unknown black-box victim systems.

| Constrained Black-box Poison-splat attack with $\epsilon = 16/255$ against Mip-Splatting | | | | | | |
|---|---|---|---|---|---|---|
| **Metric** | **Number of Gaussians** | | **Peak GPU memory [MB]** | | **Training time [minutes]** | |
| Setting / Scene | clean | poisoned | clean | poisoned | clean | poisoned |
| NS-chair | 0.372 M | 1.088 M (2.92x↑) | 6318 | **33042**\* (5.23x↑) | 7.85 | 13.75 (1.75x↑) |
| NS-drums | 0.429 M | 0.773 M (1.80x↑) | 7474 | 22036 (2.95x↑) | 8.15 | 12.60 (1.55x↑) |
| NS-ficus | 0.234 M | 0.420 M (1.79x↑) | 6144 | 12374 (2.01x↑) | 6.35 | 8.97 (1.41x↑) |
| NS-hotdog | 0.200 M | 1.437 M (7.18x↑) | 5350 | **58878**\* (11.01x↑) | 7.08 | 20.08 (2.84x↑) |
| NS-lego | 0.307 M | 1.127 M (3.67x↑) | 6696 | **48682**\* (7.27x↑) | 7.28 | 16.43 (2.26x↑) |
| NS-materials | 0.216 M | 0.505 M (2.34x↑) | 5278 | 9674 (1.83x↑) | 6.72 | 8.88 (1.32x↑) |
| NS-mic | 0.268 M | 0.557 M (2.08x↑) | 5338 | 19758 (3.70x↑) | 9.12 | 12.23 (1.34x↑) |
| NS-ship | 0.330 M | 1.793 M (5.43x↑) | 6500 | **61026**\* (9.39x↑) | 8.75 | 21.05 (2.41x↑) |
| MIP-bicycle | 8.683 M | **DoS** | 80614 | **DoS** | 47.57 | **DoS** |
| MIP-bonsai | 1.670 M | 13.016 M (7.79x↑) | 30876 | **80826**\* (2.62x↑) | 23.72 | 61.82 (2.61x↑) |
| MIP-counter | 1.493 M | 8.329 M (5.58x↑) | 19478 | **79904**\* (4.10x↑) | 26.25 | 56.75 (2.16x↑) |
| MIP-flowers | 3.834 M | 7.281 M (1.90x↑) | 50922 | **80286**\* (1.58x↑) | 32.85 | 41.80 (1.27x↑) |
| MIP-garden | 5.828 M | 8.677 M (1.49x↑) | 70446 | **80440**\* (1.14x↑) | 43.20 | 54.43 (1.26x↑) |
| MIP-kitchen | 2.182 M | 10.734 M (4.92x↑) | 37024 | **81006**\* (2.19x↑) | 30.63 | 66.92 (2.18x↑) |
| MIP-room | 2.080 M | 12.949 M (6.23x↑) | 31616 | **81130**\* (2.57x↑) | 27.13 | 77.83 (2.87x↑) |
| MIP-stump | 5.920 M | 13.925 M (2.35x↑) | 65882 | **80480**\* (1.22x↑) | 34.33 | 56.43 (1.64x↑) |
| **Unconstrained Black-box Poison-splat attack with $\epsilon = \infty$ against Mip-Splatting** | | | | | | |
| **Metric** | **Number of Gaussians** | | **Peak GPU memory [MB]** | | **Training time [minutes]** | |
| Setting / Scene | clean | poisoned | clean | poisoned | clean | poisoned |
| NS-chair | 0.372 M | 6.106 M (16.41x↑) | 6318 | **80732**\* (12.78x↑) | 7.85 | 57.73 (7.35x↑) |
| NS-drums | 0.429 M | 6.818 M (15.89x↑) | 7474 | **81210**\* (10.87x↑) | 8.15 | 67.67 (8.30x↑) |
| NS-ficus | 0.234 M | 4.847 M (20.71x↑) | 6144 | **80834**\* (13.16x↑) | 6.35 | 45.85 (7.22x↑) |
| NS-hotdog | 0.200 M | 6.876 M (34.38x↑) | 5350 | **80630**\* (15.07x↑) | 7.08 | 59.80 (8.45x↑) |
| NS-lego | 0.307 M | 6.472 M (21.08x↑) | 6696 | **80668**\* (12.05x↑) | 7.28 | 64.93 (8.92x↑) |
| NS-materials | 0.216 M | 5.513 M (25.52x↑) | 5278 | **81112**\* (15.37x↑) | 6.72 | 45.08 (6.71x↑) |
| NS-mic | 0.268 M | 5.433 M (20.27x↑) | 5338 | **81020**\* (15.18x↑) | 9.12 | 52.83 (5.79x↑) |
| NS-ship | 0.330 M | 6.848 M (20.75x↑) | 6500 | **81236**\* (12.50x↑) | 8.75 | 66.15 (7.56x↑) |
| MIP-bicycle | 8.683 M | **DoS** | 80614 | **DoS** | 47.57 | **DoS** |
| MIP-bonsai | 1.670 M | **DoS** | 30876 | **DoS** | 23.72 | **DoS** |
| MIP-counter | 1.493 M | **DoS** | 19478 | **DoS** | 26.25 | **DoS** |
| MIP-flowers | 3.834 M | **DoS** | 50922 | **DoS** | 32.85 | **DoS** |
| MIP-garden | 5.828 M | **DoS** | 70446 | **DoS** | 43.20 | **DoS** |
| MIP-kitchen | 2.182 M | **DoS** | 37024 | **DoS** | 30.63 | **DoS** |
| MIP-room | 2.080 M | **DoS** | 31616 | **DoS** | 27.13 | **DoS** |
| MIP-stump | 5.920 M | **DoS** | 65882 | **DoS** | 34.33 | **DoS** |

## E  NOTATIONS

We provide Table 7 to list all the notations appeared in the paper.

## F  VISUALIZATIONS

In this section, we present both the attacker-crafted poisoned images and the resultant reconstructions by the victim model. The visualizations aim to provide readers with more straightforward understanding of our attack. The constrained attack visualizations are provided in Table 8, and the unconstrained attack visualizations are provided in Table 9.

We have following interesting observations:

Table 7: Notation List

| Variable | Description |
| --- | --- |
| $\mathcal{G}$ | 3D Gaussians |
| $\|\mathcal{G}\|$ | The number of 3D Gaussians |
| $G_i$ | The $i$-th Gaussian in the scene |
| $\mathcal{D}$ | Clean image datasets |
| $\mathcal{D}_p$ | Poisoned image datasets |
| $V_k$ | The $k$-th camera view's clean image |
| $P_k$ | The $k$-th camera view's camera pose |
| $\tilde{V}_k$ | The $k$-th camera view's poisoned image |
| $\tilde{P}_k$ | The $k$-th camera view's poisoned pose |
| $V'_k$ | The $k$-th camera view's rendered image |
| $N$ | The total number of views |
| $\mu \in \mathbb{R}^3$ | Center 3D coordinates (positions) of a Gaussian |
| $s \in \mathbb{R}^3$ | 3D scales of a Gaussian |
| $\alpha \in \mathbb{R}$ | Opacity of a Gaussian |
| $q \in \mathbb{R}^4$ | Rotation quaternion of a Gaussian |
| $c \in \mathbb{R}^{3 \times (d+1)^2}$ | Color of a Gaussian represented as spherical harmonics, where $d$ is the degree |
| $\tau_g$ | Gradient threshold |
| $\tau_s$ | Scale threshold |
| $\tau_\alpha$ | Opacity threshold |
| $\mathcal{R}$ | Renderer function |
| $\epsilon$ | Perturbation range |
| $\eta$ | Perturbation step size |
| $T$ | The iteration number of inner optimization |
| $\tilde{T}$ | The iteration number of outer optimization |
| $\mathcal{P}_\epsilon(\cdot)$ | Projection into $\epsilon$-ball |
| $\mathcal{L}$ | Reconstruction loss |
| $\mathcal{L}_1$ | $\mathcal{L}_1$ loss |
| $\mathcal{L}_{\text{D-SSIM}}$ | Structural similarity index measure (SSIM) loss |
| $\mathcal{C}(\mathcal{G}^*)$ | Training costs of 3D Gaussians |
| $\mathcal{S}_{\text{TV}}(V_k)$ | Total variation score |
| $\lambda$ | Loss trade-off parameters |

- **Unconstrained Poisoned Attacks**: On both NeRF-Synthetic and MIP-NeRF360, the reconstruction results consistently show a low PSNR (around 20 dB). While the poisoned images and victim reconstructions may both resemble chaotic arrangements of lines with needle-like Gaussians, subtle differences in detail exist. These differences, though difficult for the human eye to distinguish, are reflected in the PSNR.

- **Constrained Attacks on NeRF-Synthetic**: the reconstruction results are significantly better than unconstrained attacks, with an average PSNR of around 30 dB.

- **Constrained Attacks on MIP-NeRF360**: There are substantial appearance differences between the victim reconstructions and the poisoned input images, which are easily noticeable through human observation. Specifically, in the bicycle scene, the reconstructed trees appear to have denser leaves. In the bonsai and counter scenes, we observe notable changes in reflective areas, with several reflection highlights disappearing in the victim's reconstructed images. Additionally, in the counter scene, there is evidence of underfitting, where the complex textures of the poisoned data degrade into large, blurry floaters in the victim's reconstruction.

Table 8: Attacker poisoned and victim reconstructed images with PSNR values under constrained attack ($\epsilon = 16/255$) settings.

| Dataset Setting | Attacker Poisoned Image | Victim Reconstructed Image | PSNR |
|---|---|---|---|
| NS-Chair |  |  | 37.07 dB |
| NS-Drums |  |  | 30.32 dB |
| NS-Ficus |  |  | 35.77 dB |
| MIP-bicycle |  |  | 18.20 dB |
| MIP-bonsai |  |  | 22.67 dB |
| MIP-counter |  |  | 24.45 dB |

Table 9: Attacker poisoned and victim reconstructed images with PSNR values under unconstrained attack settings.

| Dataset Scene | Attacker Poisoned Image | Victim Reconstructed Image | PSNR |
|---|---|---|---|
| NS-Chair | | | 19.54 dB |
| NS-Drums | | | 18.65 dB |
| NS-Ficus | | | 21.00 dB |
| MIP-bicycle | | | 20.57 dB |
| MIP-bonsai | | | 23.66 dB |
| MIP-counter | | | 21.33 dB |

Table 10: Attack performance (number of Gaussians, GPU memory consumption and training time) and image quality (SSIM and PSNR compared with clean images) under different levels of perturbation range on NeRF Synthetic dataset.

| Scene | Attack setting | Number of Gaussians | GPU memory | Training time | SSIM | PSNR |
|---|---|---|---|---|---|---|
| chair | clean | 0.494 M | 3752 MB | 9.45 min | - | - |
| | $\epsilon = 8/255$ | 0.670 M | 4520 MB | 11.24 min | 0.42 | 34.20 |
| | $\epsilon = 16/255$ | 0.957 M | 8003 MB | 13.41 min | 0.21 | 27.94 |
| | $\epsilon = 24/255$ | 1.253 M | 16394 MB | 15.89 min | 0.16 | 24.35 |
| | unconstrained | 4.152 M | 51424 MB | 40.71 min | 0.07 | 4.63 |
| drums | clean | 0.385 M | 3456 MB | 8.75 min | - | - |
| | $\epsilon = 8/255$ | 0.469 M | 4353 MB | 10.19 min | 0.45 | 34.05 |
| | $\epsilon = 16/255$ | 0.695 M | 7190 MB | 11.94 min | 0.24 | 27.87 |
| | $\epsilon = 24/255$ | 0.966 M | 14668 MB | 14.86 min | 0.17 | 24.36 |
| | unconstrained | 4.109 M | 62328 MB | 43.25 min | 0.03 | 4.49 |
| ficus | clean | 0.265 M | 3194 MB | 6.99 min | - | - |
| | $\epsilon = 8/255$ | 0.246 M | 3238 MB | 8.28 min | 0.39 | 34.45 |
| | $\epsilon = 16/255$ | 0.273 M | 3783 MB | 8.94 min | 0.18 | 28.21 |
| | $\epsilon = 24/255$ | 0.450 M | 8993 MB | 11.32 min | 0.13 | 24.53 |
| | unconstrained | 3.718 M | 50450 MB | 36.92 min | 0.08 | 4.41 |
| hotdog | clean | 0.185 M | 3336 MB | 8.57 min | - | - |
| | $\epsilon = 8/255$ | 0.610 M | 10362 MB | 12.96 min | 0.46 | 33.93 |
| | $\epsilon = 16/255$ | 1.147 M | 29747 MB | 17.43 min | 0.25 | 27.56 |
| | $\epsilon = 24/255$ | 1.553 M | 39087 MB | 21.26 min | 0.17 | 24.11 |
| | unconstrained | 4.272 M | 47859 MB | 38.85 min | 0.07 | 5.75 |
| lego | clean | 0.341 M | 3532 MB | 8.62 min | - | - |
| | $\epsilon = 8/255$ | 0.469 M | 4648 MB | 10.38 min | 0.48 | 34.08 |
| | $\epsilon = 16/255$ | 0.805 M | 9960 MB | 13.02 min | 0.28 | 27.88 |
| | $\epsilon = 24/255$ | 1.116 M | 18643 MB | 15.33 min | 0.21 | 24.30 |
| | unconstrained | 4.159 M | 78852 MB | 42.46 min | 0.06 | 4.78 |
| materials | clean | 0.169 M | 3172 MB | 7.33 min | - | - |
| | $\epsilon = 8/255$ | 0.229 M | 3581 MB | 8.84 min | 0.44 | 34.25 |
| | $\epsilon = 16/255$ | 0.410 M | 4886 MB | 9.92 min | 0.21 | 27.97 |
| | $\epsilon = 24/255$ | 0.589 M | 7724 MB | 11.51 min | 0.14 | 24.35 |
| | unconstrained | 3.380 M | 43998 MB | 32.40 min | 0.08 | 5.40 |
| mic | clean | 0.205 M | 3499 MB | 8.08 min | - | - |
| | $\epsilon = 8/255$ | 0.251 M | 4038 MB | 9.29 min | 0.36 | 34.46 |
| | $\epsilon = 16/255$ | 0.359 M | 6437 MB | 10.96 min | 0.15 | 28.17 |
| | $\epsilon = 24/255$ | 0.514 M | 13014 MB | 12.89 min | 0.10 | 24.48 |
| | unconstrained | 3.940 M | 61835 MB | 39.02 min | 0.08 | 4.50 |
| ship | clean | 0.272 M | 3692 MB | 8.87 min | - | - |
| | $\epsilon = 8/255$ | 0.516 M | 5574 MB | 11.01 min | 0.55 | 33.42 |
| | $\epsilon = 16/255$ | 1.071 M | 16666 MB | 16.12 min | 0.33 | 27.50 |
| | $\epsilon = 24/255$ | 1.365 M | 29828 MB | 18.46 min | 0.24 | 24.03 |
| | unconstrained | 4.317 M | 80956 MB | 44.11 min | 0.04 | 5.31 |

# G ABLATION STUDY ON PERTURBATION RANGE

We compare the attack effects (in terms of number of Gaussians, GPU memory and training time) and image degradation (in terms of SSIM and PSNR compared with clean input) under different perturbation ranges $\epsilon = 8/255$, $\epsilon = 16/255$, $\epsilon = 24/255$) on NeRF-Synthetic and MIP-NeRF360 datasets, as shown in Table 10 and Table 11.

Table 11: Attack performance (number of Gaussians, GPU memory consumption and training time) and image quality (SSIM and PSNR compared with clean images) under different levels of constraints on MIP-NeRF360 dataset.

| Scene | Attack setting | Number of Gaussians | GPU memory (MB) | Training time (min) | SSIM | PSNR |
|---|---|---|---|---|---|---|
| bicycle | clean | 5.793 M | 17748 | 33.42 | - | - |
| | $\epsilon = 8/255$ | 6.116 M | 18608 | 34.73 | 0.86 | 30.80 |
| | $\epsilon = 16/255$ | 10.129 M | 27074 | 44.44 | 0.67 | 26.24 |
| | $\epsilon = 24/255$ | 13.265 M | 34870 | 51.37 | 0.52 | 23.12 |
| | unconstrained | 25.268 M | 63236 | 81.48 | 0.02 | 6.60 |
| bonsai | clean | 1.294 M | 10216 | 21.68 | - | - |
| | $\epsilon = 8/255$ | 1.778 M | 11495 | 21.39 | 0.79 | 32.45 |
| | $\epsilon = 16/255$ | 6.150 M | 20564 | 32.51 | 0.51 | 26.83 |
| | $\epsilon = 24/255$ | 9.321 M | 27191 | 39.84 | 0.35 | 23.52 |
| | unconstrained | 20.127 M | 54506 | 75.18 | 0.01 | 6.27 |
| counter | clean | 1.195 M | 10750 | 26.46 | - | - |
| | $\epsilon = 8/255$ | 1.739 M | 15133 | 28.06 | 0.80 | 32.24 |
| | $\epsilon = 16/255$ | 4.628 M | 29796 | 37.63 | 0.52 | 26.78 |
| | $\epsilon = 24/255$ | 6.649 M | 47607 | 43.68 | 0.35 | 23.45 |
| | unconstrained | 11.167 M | 80732 | 62.04 | 0.01 | 6.64 |
| flowers | clean | 3.508 M | 12520 | 25.47 | - | - |
| | $\epsilon = 8/255$ | 3.457 M | 12391 | 25.53 | 0.87 | 29.18 |
| | $\epsilon = 16/255$ | 5.177 M | 16338 | 30.05 | 0.71 | 25.60 |
| | $\epsilon = 24/255$ | 7.280 M | 20264 | 34.25 | 0.58 | 22.79 |
| | unconstrained | 18.075 M | 45515 | 62.62 | 0.02 | 6.79 |
| garden | clean | 5.684 M | 17561 | 33.19 | - | - |
| | $\epsilon = 8/255$ | 5.122 M | 16498 | 32.17 | 0.84 | 29.85 |
| | $\epsilon = 16/255$ | 7.697 M | 23115 | 40.82 | 0.68 | 25.87 |
| | $\epsilon = 24/255$ | 10.356 M | 28130 | 48.88 | 0.54 | 22.82 |
| | unconstrained | 21.527 M | 52140 | 83.81 | 0.04 | 7.43 |
| kitchen | clean | 1.799 M | 12442 | 26.61 | - | - |
| | $\epsilon = 8/255$ | 2.675 M | 13760 | 31.39 | 0.83 | 31.58 |
| | $\epsilon = 16/255$ | 6.346 M | 20902 | 43.24 | 0.60 | 26.52 |
| | $\epsilon = 24/255$ | 9.445 M | 27646 | 53.16 | 0.45 | 23.18 |
| | unconstrained | 12.830 M | 77141 | 73.04 | 0.03 | 7.76 |
| room | clean | 1.513 M | 12316 | 25.36 | - | - |
| | $\epsilon = 8/255$ | 3.034 M | 19542 | 32.99 | 0.75 | 32.41 |
| | $\epsilon = 16/255$ | 7.129 M | 46238 | 47.98 | 0.45 | 26.64 |
| | $\epsilon = 24/255$ | 9.724 M | 80142 | 56.87 | 0.29 | 23.32 |
| | unconstrained | 16.019 M | 57540 | 76.25 | 0.01 | 6.42 |
| stump | clean | 4.671 M | 14135 | 27.06 | - | - |
| | $\epsilon = 8/255$ | 5.152 M | 15342 | 28.21 | 0.84 | 31.20 |
| | $\epsilon = 16/255$ | 10.003 M | 25714 | 38.48 | 0.65 | 25.67 |
| | $\epsilon = 24/255$ | 14.292 M | 34246 | 46.89 | 0.49 | 22.98 |
| | unconstrained | 13.550 M | 36181 | 51.51 | 0.02 | 7.14 |

## H    ABLATION STUDY ON POISON RATIO

In the main paper, we conducted all other experiments with 100% poison rate, based on the assumption that attacker can fully control data collection process. This assumption is reasonable for most 3D tasks, since the datasets typically are the same scene or object captured from different angles.

To gain deeper insights into the Poison-splat attack, we explore how varying levels of data poisoning ratio can affect the attack's effectiveness. We conduct experiments using poison ratios of 20%, 40%, 60%, and 80%, and report the number of Gaussians, GPU memory consumption, and training time in Table 12. In these experiments, the poisoned views are randomly selected. To mitigate the effects of randomness, we report the mean value and standard deviation for each setting across three individual runs.

Table 12: Attack effectiveness under different poison ratio.

| Scene | Poison Ratio | Number of Gaussians | GPU memory (MB) | Training time (min) |
|---|---|---|---|---|
| NS-hotdog-eps16 | clean | 0.185 M $\pm$ 0.000 M | 3336 MB $\pm$ 11 MB | 8.57 min $\pm$ 0.30 min |
| | 20% | 0.203 M $\pm$ 0.004 M | 3286 MB $\pm$ 70 MB | 8.84 min $\pm$ 0.27 min |
| | 40% | 0.279 M $\pm$ 0.024 M | 3896 MB $\pm$ 286 MB | 9.79 min $\pm$ 0.53 min |
| | 60% | 0.501 M $\pm$ 0.054 M | 7367 MB $\pm$ 1200 MB | 11.33 min $\pm$ 0.82 min |
| | 80% | 0.806 M $\pm$ 0.018 M | 13621 MB $\pm$ 861 MB | 14.02 min $\pm$ 0.60 min |
| | 100% | 1.147 M $\pm$ 0.003 M | 29747 $\pm$ 57 MB | 17.43 min $\pm$ 0.61 min |
| NS-ship-eps16 | clean | 0.272 M $\pm$ 0.001 M | 3692 MB $\pm$ 52 MB | 8.87 min $\pm$ 0.44 min |
| | 20% | 0.321 M $\pm$ 0.004 M | 3764 MB $\pm$ 70 MB | 9.55 min $\pm$ 0.17 min |
| | 40% | 0.472 M $\pm$ 0.021 M | 4796 MB $\pm$ 293 MB | 10.49 min $\pm$ 0.58 min |
| | 60% | 0.718 M $\pm$ 0.003 M | 7034 MB $\pm$ 407 MB | 12.52 min $\pm$ 0.45 min |
| | 80% | 0.924 M $\pm$ 0.004 M | 11850 MB $\pm$ 510 MB | 14.37 min $\pm$ 0.60 min |
| | 100% | 1.071 M $\pm$ 0.003 M | 16666 MB $\pm$ 548 MB | 16.12 min $\pm$ 0.42 min |
| MIP-counter-eps16 | clean | 1.195 M $\pm$ 0.005 M | 10750 MB $\pm$ 104 MB | 26.46 min $\pm$ 0.57 min |
| | 20% | 1.221 M $\pm$ 0.005 M | 11043 MB $\pm$ 141 MB | 27.41 min $\pm$ 0.85 min |
| | 40% | 1.358 M $\pm$ 0.030 M | 11535 MB $\pm$ 147 MB | 27.97 min $\pm$ 0.47 min |
| | 60% | 2.005 M $\pm$ 0.056 M | 13167 MB $\pm$ 227 MB | 28.87 min $\pm$ 0.52 min |
| | 80% | 3.273 M $\pm$ 0.119 M | 19578 MB $\pm$ 1417 MB | 32.99 min $\pm$ 0.57 min |
| | 100% | 4.628 M $\pm$ 0.014 M | 29796 MB $\pm$ 187 MB | 37.63 min $\pm$ 0.48 min |
| MIP-room-eps16 | clean | 1.513 M $\pm$ 0.004 M | 12316 MB $\pm$ 13 MB | 25.36 min $\pm$ 0.09 min |
| | 20% | 1.520 M $\pm$ 0.015 M | 12624 MB $\pm$ 502 MB | 26.45 min $\pm$ 0.75 min |
| | 40% | 1.938 M $\pm$ 0.036 M | 14495 MB $\pm$ 320 MB | 27.99 min $\pm$ 0.72 min |
| | 60% | 3.589 M $\pm$ 0.096 M | 18760 MB $\pm$ 765 MB | 34.09 min $\pm$ 0.79 min |
| | 80% | 5.503 M $\pm$ 0.017 M | 28166 MB $\pm$ 457 MB | 41.47 min $\pm$ 1.36 min |
| | 100% | 7.129 M $\pm$ 0.020 M | 46238 MB $\pm$ 802 MB | 47.98 min $\pm$ 0.72 min |

From Table 12, we have following observations:

- Attack performance, in terms of both GPU memory usage and training time extension, becomes stronger as poisoning rate increases.
- At higher poisoning rates (between 60%-80%), the attack exhibits greater variance due to the randomness involved in selecting which views to poison. This suggests that the choice of views to poison can also impact the effectiveness of the attack.

## I    ABLATION STUDY ON DEFENSE THRESHOLD OF LIMITING GAUSSIAN NUMBERS

In the main paper, we set the defense threshold to ensure a tight wrapping of necessary Gaussians for reconstruction on the clean scene. More specifically, for three scenes we show in Section 4.4 and Figure 5, we set defense threshold as in the following Table 13:

For a 3DGS trainer, setting a uniform defense threshold is a notably challenging task due to the significant variance in different scenes. Implementing a low threshold can safeguard the victim's computational resources from excessive consumption, but it may also substantially compromise

Table 13: Number of Gaussians on clean data and as defense threshold used in Section 4.4.

| Scene | #Gaussians on clean data | #Gaussians as defense threshold |
|---|---|---|
| Bicycle | 5.793 Million | 6 Million |
| Flowers | 3.508 Million | 4 Million |
| Room | 1.513 Million | 2 Million |

the quality of reconstruction for complex scenes. Choosing the defense threshold should consider this resource-quality tradeoff. Taking *room* of Nerf-Synthetic as an example, we tested the defense threshold acrosss 2 million to 7 million, as shown in the following Table 14 and Figure 6:

Table 14: Defense threshold impact on GPU consumption and reconstruction PSNR.

| Defense threshold | Max GPU memory (MB) | Reconstruction PSNR |
|---|---|---|
| *Clean input* | 12316 MB | 29.21 |
| 2 Million | 14192 MB | 19.03 |
| 3 Million | 16472 MB | 22.27 |
| 4 Million | 18784 MB | 24.11 |
| 5 Million | 23642 MB | 24.49 |
| 6 Million | 36214 MB | 28.02 |
| *Poison + No Defense* | 46238 MB | 29.08 |

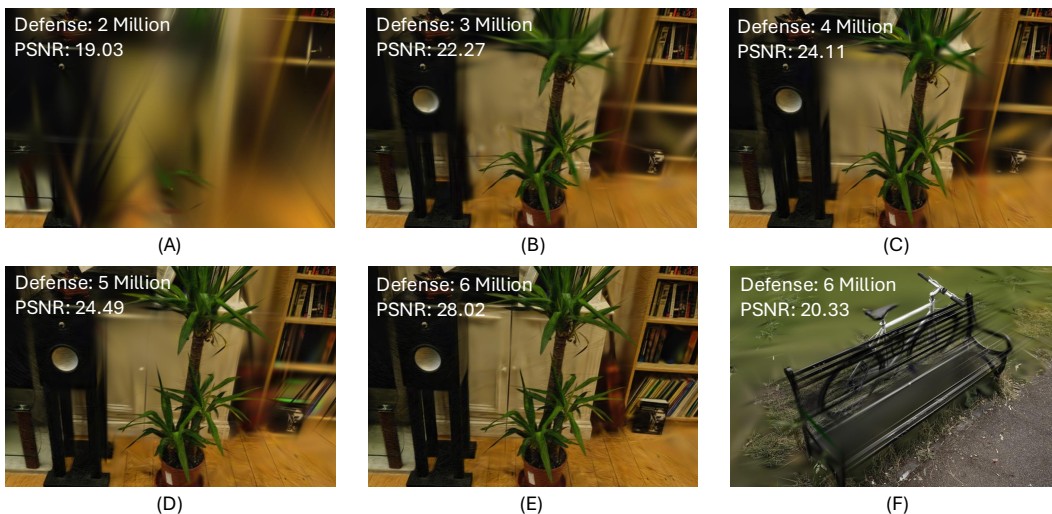

Figure 6: Rendering of reconstructed models under different defense threshold.

As the defense threshold becomes tighter, the maximum GPU memory usage is confined within a safer range. However, this results in a significant drop in the reconstruction's PSNR, falling to as low as 19.03, which nearly makes it unusable. (c.f. Figure 6 (A)). What's more, while setting a threshold at 6 million Gaussians appears adequate (achieving PSNR 28.02 dB), this same threshold yields poor rendering results for *bicycle* scene, with only 20.33 dB PSNR (c.f. Figure 6 (F)).

Overall, determining a universal defense threshold that preserves reconstruction quality across various scenes is challenging. Designing a more effective defense against computation cost attack remains an open question, and we hope that future research will continue to explore this area.

## J    IMAGE SMOOTHING IS NOT AN IDEAL DEFENSE

We will show that image smoothing as a pre-processing to Gaussian Splatting training procedure is not an ideal defense. Without a reliable detection method, defenders may only resort to universally

applying image smoothing to all incoming data. This pre-processing will significantly compromise reconstruction quality.

We found that although image smoothing may reduce GPU consumption to some extent, it severly undermines efforts to preserve fine image details (Yu et al., 2024; Ye et al., 2024; Yan et al., 2024b). As illustrated in Figure 7, applying common smoothing techniques such as Gaussian filtering or Bilateral filtering (Tomasi & Manduchi, 1998) leads to a substantial degradation in reconstruction quality. For instance, on the *chair* scene of NeRF-Synthetic dataset, reconstruction achieves 36.91 dB PSNR without pre-processing; however, with Gaussian or Bilateral filtering, the PSNR drops sharply to around 25 dB. This level of degradation is clearly undesirable. Given these challenges, we urge the community to develop more sophiscated detection and defense mechanisms against computation cost attacks, balancing resource consumption with the preservation of image quality.

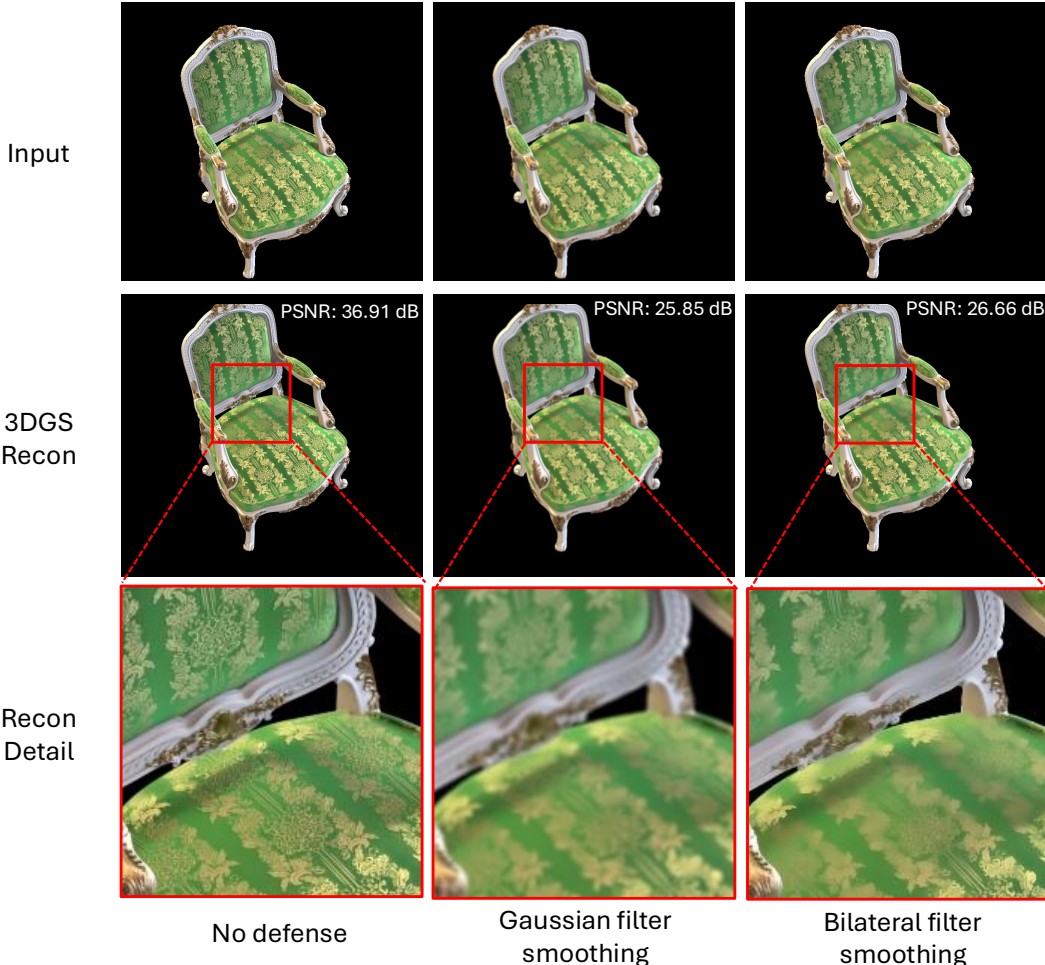

Figure 7: Using image smoothing as a pre-processing step can result in unsatisfactory model reconstruction quality, particularly regarding fine details in images. This naive defense strategy significantly degrades the PSNR by approximately 10 dB, contradicting the primary goal of the service provider to maintain high-quality outputs.

## K    TIME EFFICIENCY OF POISON-SPLAT ATTACK

According to Algorithm 1, we need to solve a bi-level optimization problem, and the time complexity is related to $T$ (inner iterations) and $\tilde{T}$ (outer iterations). However in practice, our inner iterations $T$ are far fewer than outer iterations $\tilde{T}$, and the actual time cost of attacker is even less than victim's training time, which is an advantage of our attack.

Specifically, the total running time of algorithm 1 is decided by two parts:

1. Train a proxy model on clean data (Line 2 of Algorithm 1);
2. Solve the bi-level optimization (Line 3 to Line 13 of Algorithm 1)

For example, for unconstrained attacks on MIP-NeRF360, we set $T = 6000$ and $\tilde{T} = 25$, and the time of attack is totally acceptable, and is even shorter than the victim training, as shown in the Table 15 below:

Table 15: Comparison of training times for the attacker and the victim.

| Scene | Attacker: Proxy training + Bi-level optimization | Victim: Training time |
|---|---|---|
| Bicycle | 33.42 + 16.57 min | 81.48 min |
| Bonsai | 21.68 + 14.00 min | 75.18 min |
| Counter | 26.46 + 15.23 min | 62.04 min |
| Flowers | 25.47 + 15.05 min | 62.62 min |
| Garden | 33.19 + 15.07 min | 83.81 min |
| Kitchen | 26.61 + 14.77 min | 73.04 min |
| Room | 25.36 + 16.50 min | 76.25 min |
| Stump | 27.06 + 15.63 min | 51.51 min |

Lastly, the primary motivation of our work is to expose the severity of this security backdoor, and attack efficiency is beyond the scope of this paper. We hope to leave the improvement of attack efficiency in future studies.

