# OpenReview forum: "Poison-splat: Computation Cost Attack on 3D Gaussian Splatting"
_ICLR.cc/2025/Conference — ICLR 2025 Spotlight_

### Official Review · Reviewer_THFm · 2024-10-29

**Soundness:** 4
**Presentation:** 4
**Contribution:** 4
**Rating:** 8
**Confidence:** 3

**Summary:**

This work reveals a major security vulnerability that has been overlooked in 3D Gaussian Splatting (3DGS): the computational cost of training 3DGS can be maliciously manipulated by poisoning the input data. This paper introduces a novel attack, termed "Poison," in which an adversary can poison the input images, thereby significantly increasing the memory and computational time required for 3DGS training, ultimately pushing the algorithm to its highest computational complexity. In extreme cases, the attack can even exhaust all available memory, leading to a denial-of-service (DoS) event on the server and causing real harm to 3DGS service providers.
The attack is modeled as a two-layer optimization problem, addressed through three strategies: attack target approximation, proxy model rendering, and optional constrained optimization. This proposed approach not only ensures the effectiveness of the attack but also makes it challenging for simple, existing defenses to succeed. This novel attack aims to raise awareness of the potential vulnerabilities in 3DGS systems.

**Strengths:**

1. It reveals a major security vulnerability that has been overlooked in 3DGS.
2. The attack's effectiveness is validated across various datasets.

**Weaknesses:**

1. It lacks evaluation of the attack's effects on the inference phase.
2. There are no intuitive metrics for evaluating the success or failure of this attack.

**Questions:**

(1) As a model training service provider, certain benchmarks, such as model size and typical training duration, are commonly understood. Given these expectations, would an attack that poisons samples to increase memory usage and training time be easy to detect? Additionally, could existing methods for detecting adversarial samples be effective in identifying these poisoned samples?
(2)Are existing attack methods targeting the inference phase applicable to the training phase? If so, could these methods be included in the experimental analysis for comparison? If not, could the authors explain why?
(3) Since this attack targets the training phase, it would be helpful if the authors could analyze the required percentage of contamination in a clean dataset for the attack to succeed.
(4) In the Experimental Analysis section, the authors conducted extensive experiments to analyze the impact of the attack on memory usage and training duration, which is commendable. However, while the abstract suggests that Poison-splat can lead to a DoS damage to the server, the Experimental Analysis section lacks any evaluation of the attack's effects on the inference phase.
(5) Could the authors add more common models to compare the effectiveness of black-box and white-box attacks?
(6) The primary objective of the attack is to consume excessive computational resources, but what are the most serious consequences of this? Is it a denial-of-service (DoS) scenario? Additionally, how difficult would it be to achieve significant damage? For instance, if an attacker aims to prevent a model from completing its training, would this require prior knowledge of the service provider's computational resources? Lastly, it is unclear how the success or failure of this type of attack should be assessed. For example, if the service provider’s computational resources are sufficient to handle the increased demand, should the attack then be considered unsuccessful?

---

> ### Author Response · Authors · 2024-11-22
> **Response to Reviewer THFm (Split 1 of 5)**
>
> We sincerely thank the reviewer for acknowledging the significance of our finding and the effectiveness of our proposed attack. In the following we address the main concerns.
>
> **W1 & Q4. While Poison-splat can lead to DoS, it lacks evaluation of inference phase.**
> >`W1. It lacks evaluation of the attack's effects on the inference phase.`
>
> >`Q4. In the Experimental Analysis section, the authors conducted extensive experiments to analyze the impact of the attack on memory usage and training duration, which is commendable. However, while the abstract suggests that Poison-splat can lead to a DoS damage to the server, the Experimental Analysis section lacks any evaluation of the attack's effects on the inference phase.`
>
> Thank you for raising this concern. First we still want to highlight that **our attack is designed particularly for the training phase**, and the denial-of-service is meant to happen in the training phase. Please refer to **Q4 of Reviewer SqKM** where denial-of-service is frequently encountered when training on RTX A5000 or RTX 4090.
>
>
> Although it is a training-phase attack in design, we can still find some consequences of the attack in the inference phase. For example, we identify the slowing down of rendering speed featured by a significantly smaller FPS (c.f. rightmost columns of Table 1, Table 3 and Table 4). We conclude the average FPS drop in the following table:
>
> | Dataset - attack setting | Average Clean Render Speed (FPS) | Average Poisoned Render Speed (FPS) | Average slowing down |
> |:---:|:---:|:---:|:---:|
> |Nerf-Synthetic-eps16 | 352.250 | 165.125| 2.13x $\downarrow$ |
> |Nerf-Synthetic-unconstrained | 352.250 | 28.250|12.47x $\downarrow$|
> |MIP-Nerf360-eps16 | 130.333 | 56.778 | 2.30x $\downarrow$|
> |MIP-Nerf360-unconstrained | 130.333 | 19.778 | 6.59x $\downarrow$|
>
>
> On the other hand, our attack can cause a higher storage overhead of trained 3DGS model during inference stage, which is self-evident as the final number of Gaussians are massively increased.
>
> | Dataset - attack setting | Average Clean Model Size (GB) | Average Poisoned Model Size (GB) | Average storage increase |
> |:---:|:---:|:---:|:---:|
> |Nerf-Synthetic-eps16 | 0.067 | 0.165 |  2.46x$\uparrow$ |
> |Nerf-Synthetic-unconstrained | 0.067 | 0.925 | 13.80x $\uparrow$|
> |MIP-Nerf360-eps16 | 0.740 | 1.615 | 2.18x $\uparrow$|
> |MIP-Nerf360-unconstrained | 0.740 | 3.906 | 5.28x $\uparrow$|
>
> ---
> **W2. No intuitive metrics for evluating attack success/failure**.
> >`W2. There are no intuitive metrics for evaluating the success or failure of this attack.`
>
> In this paper, we consider the computation cost attack successful if the resulting computation cost exceeds that required for clean input. If the attacker wishes to apply higher standards to measure attack effectiveness, they can establish metrics based on scales of the original computation cost.
>
>
> For example, we can identify the GPU memory usage exceeding 100%, 150% and 200% as three levels of successful attack states. The attack success rate on these three levels can be concluded in the following table:
>
>
> | Attack Dataset - setting | ASR@100%GPU | ASR@150%GPU | ASR@200%GPU |
> |:--|:--:|:--:|:--:|
> | NeRF Synthetic  $\epsilon = 16/255$ | 100% | 87.5%| 62.5%|
> | MIP-NeRF360 $\epsilon = 16/255$ | 100% | 77.8% | 33.3% |
> | NeRF Synthetic unconstrained | 100% | 100% | 100%  |
> | MIP-NeRF360 unconstrained | 100% | 100% | 100% |

---

> ### Author Response · Authors · 2024-11-22
> **Response to Reviewer THFm (Split 2 of 5)**
>
> **Q1.(1) Training provider has expectations to certain benchmark, would it be easy to detect?**
> >`Q1(1). As a model training service provider, certain benchmarks, such as model size and typical training duration, are commonly understood. Given these expectations, would an attack that poisons samples to increase memory usage and training time be easy to detect?`
>
> For the victim (online 3DGS service providers), the data they are dealing with are in-the-wild unseen data from various real-world customers, not from well-known, public benchmarks. In such real-world scenarios, the service provider actually has no clue to predict precise training cost in advance.
>
> We tested our attack on a real-world 3DGS training provider platform, Polycam (https://poly.cam/tools/gaussian-splatting). We uploaded our poisoned images of `room` scene in MIP-NeRF360 dataset which is a public benchmark. However we didn't encounter effective detections for these poisoned data. The attack managed to increase the output 3D file size as well as service response time, and even triggered service failure.
>
> | Number of Views | Clean file size / response time | Poisoned file size / response time |
> |:--:|:--:|:--:|
> |20 | 15.8 MB / 16 min | 31.5 MB / 24 min|
> |60 | 29.0 MB / 29 min | 46.1 MB / 56 min|
> |70 | 32.4 MB / 33 min  |**Service failed** |
> |100| 40.2 MB / 45 min | **Service failed** |
>
> ---
> **Q1(2). Will adversarial sample detection methods be effective in detecting our poisoned samples?**
> >`Q1(2). Additionally, could existing methods for detecting adversarial samples be effective in identifying these poisoned samples?`
>
> Many adversarial sample detection methods are heavily built on properties of classifiers, including those based on uncertainty/confidence[1][2][10], softmax/logits[3][4][5][6], gradients[11], or feature statistics[7][8][9]. These methods are specially designed for classification tasks and it is not straightforward to adapt to our scenario, where our victim model is a 3D generative model.
>
> For other adversarial detection not relying on classification, for example methods based on natural image/training distribution statistics[12][13][14] or denoiser methods[15][16], their generalization ability to new data and their effectiveness against adaptive attacks are worth questioning and need further in-depth analysis. We leave the exploration of effective and generalizable detection methods for future work.
>
> ***Reference***
>
> *[1] Feinman, Reuben, et al. "Detecting adversarial samples from artifacts." arXiv preprint arXiv:1703.00410 (2017).*
>
> *[2] Smith, Lewis, and Yarin Gal. "Understanding measures of uncertainty for adversarial example detection." UAI 2018.*
>
> *[3] Hendrycks, Dan, and Kevin Gimpel. "A Baseline for Detecting Misclassified and Out-of-Distribution Examples in Neural Networks." ICLR 2022.*
>
> *[4] Aigrain, Jonathan, and Marcin Detyniecki. "Detecting adversarial examples and other misclassifications in neural networks by introspection." arXiv preprint arXiv:1905.09186 (2019).*
>
> *[5] Monteiro, João, et al. "Generalizable adversarial examples detection based on bi-model decision mismatch." 2019 IEEE International Conference on Systems, Man and Cybernetics (SMC). IEEE, 2019.*
>
> *[6] Pang, Tianyu, et al. "Towards robust detection of adversarial examples." NeurIPS 2018.*
>
> *[7] Cohen, Gilad, Guillermo Sapiro, and Raja Giryes. "Detecting adversarial samples using influence functions and nearest neighbors." CVPR 2020.*
>
> *[8] Mao, Xiaofeng, et al. "Learning to characterize adversarial subspaces." ICASSP 2020.*
>
> *[9] Lu, Jiajun, Theerasit Issaranon, and David Forsyth. "Safetynet: Detecting and rejecting adversarial examples robustly." ICCV 2017.*
>
> *[10] Sotgiu, Angelo, et al. "Deep neural rejection against adversarial examples." EURASIP Journal on Information Security 2020 (2020): 1-10.*
>
> *[11] Lust, Julia, and Alexandru Paul Condurache. "GraN: an efficient gradient-norm based detector for adversarial and misclassified examples." arXiv preprint arXiv:2004.09179 (2020).*
>
> *[12] Kherchouche, Anouar, et al. "Detection of adversarial examples in deep neural networks with natural scene statistics." 2020 International Joint Conference on Neural Networks (IJCNN). IEEE, 2020.*
>
> *[13] Grosse, Kathrin, et al. "On the (statistical) detection of adversarial examples." arXiv preprint arXiv:1702.06280 (2017).*
>
> *[14] Lee, Kimin, et al. "A simple unified framework for detecting out-of-distribution samples and adversarial attacks." NeurIPS 2018.*
>
> *[15] Meng, Dongyu, and Hao Chen. "Magnet: a two-pronged defense against adversarial examples." Proceedings of the 2017 ACM SIGSAC conference on computer and communications security. 2017.*
>
> *[16] Song, Yang, et al. "PixelDefend: Leveraging Generative Models to Understand and Defend against Adversarial Examples." ICLR 2018.*

---

> ### Author Response · Authors · 2024-11-22
> **Response to Reviewer THFm (Split 3 of 5)**
>
> **Q2. Attacking training phase or attacking inference phase? Are existing inference-phase attacks applicable to training phase?**
> >`Q2. Are existing attack methods targeting the inference phase applicable to the training phase? If so, could these methods be included in the experimental analysis for comparison? If not, could the authors explain why?`
>
> We need to clarify our attack is designed for attacking training phase of 3DGS. For further clarity, we provide a detailed description outlining the input/output of training / inference phases:
>
> + **Training phase of 3DGS**: the model takes camera positions as input and outputs reconstructed view images. These reconstructed images are then compared with ground-truth view images to compute the reconstruction loss, which is subsequently used to update the model.
>
> + **Inference phase of 3DGS**: the model only receives camera positions as input, but this time, it simply outputs the view image without any learning or updating process.
>
> It is important to note that 3DGS does not have image inputs during inference. The only inputs are camera positions, which consist of a few extrinsic parameters. This setup is distinctly different from that of classifiers or neural networks. Consequently, in the inference stage, there are limited room to conduct attacks over 3DGS model.
>
> For existing inference-stage attacks, most of them exploit model's limited generalization ability (e.g. adversarial attacks, jailbreak attacks) or leverage model's behavior differences across various data distribution areas (e.g. membership inference attacks, energy-latency attacks on language models). These attacks are specifically designed for networks that are already trained and widely deployed. Their goal is not to disrupt model training process itself, but to manipulate or degrade their performance during deployment. Therefore, directly applying these inference-based attacks to training phase of 3DGS is infeasible.
>
> ---
> **Q3. Required percentage of contamination to succeed.**
> >`Q3. Since this attack targets the training phase, it would be helpful if the authors could analyze the required percentage of contamination in a clean dataset for the attack to succeed. `
>
> Thank you for raising this question. In the main paper, we only consider 100% poison rate based on the assumption that attacker can fully control the data collection process, since 3D datasets are typically composed of images of the same scene from different angles.
>
> To answer your question, we further explore how varying poisoning ratio can affect the attack’s effectiveness, as reported in the following table:
> | Scene             | Poison Ratio | Number of Gaussians | GPU memory (MB)  | Training time (min) |
> |:--|:--:|:--:|:--:|:--:|
> | **NS-hotdog-eps16** | clean        | 0.185 M ± 0.000 M   | 3336 MB ± 11 MB  | 8.57 min ± 0.30 min |
> |                   | 20%          | 0.203 M ± 0.004 M   | 3286 MB ± 70 MB  | 8.84 min ± 0.27 min |
> |                   | 40%          | 0.279 M ± 0.024 M   | 3896 MB ± 286 MB | 9.79 min ± 0.53 min |
> |                   | 60%          | 0.501 M ± 0.054 M   | 7367 MB ± 1200 MB| 11.33 min ± 0.82 min|
> |                   | 80%          | 0.806 M ± 0.018 M   | 13621 MB ± 861 MB| 14.02 min ± 0.60 min|
> |                   | 100%         | 1.147 M ± 0.003 M   | 29747 MB ± 57 MB | 17.43 min ± 0.61 min|
> |------|----|--|---|-----|
> | **MIP-counter-eps16**| clean        | 1.195 M ± 0.005 M     | 10750 MB ± 104 MB    | 26.46 min ± 0.57 min     |
> |                      | 20%          | 1.221 M ± 0.005 M     | 11043 MB ± 141 MB    | 27.41 min ± 0.85 min     |
> |                      | 40%          | 1.358 M ± 0.030 M     | 11535 MB ± 147 MB    | 27.97 min ± 0.47 min     |
> |                      | 60%          | 2.005 M ± 0.056 M     | 13167 MB ± 227 MB    | 28.87 min ± 0.52 min     |
> |                      | 80%          | 3.273 M ± 0.119 M     | 19578 MB ± 1417 MB   | 32.99 min ± 0.57 min     |
> |                      | 100%         | 4.628 M ± 0.014 M     | 29796 MB ± 187 MB    | 37.63 min ± 0.48 min     |
>
> It is evident that attack performance, in terms of both GPU memory usage and training time extension, becomes stronger as poisoning rate increases. Here we randomly selected the poisoned views and report the standard deviation across three individual runs. Please refer to **Table 12 in Appendix Section H** to see the full results.

---

> ### Author Response · Authors · 2024-11-22
> **Response to Reviewer THFm (Split 4 of 5)**
>
> **Q5. Could authors add more common models to test effectiveness of black-box and white-box attacks?**
> >`Q5. Could the authors add more common models to compare the effectiveness of black-box and white-box attacks?`
>
> Sure. In the initial paper we have provided white-box attacks for 3D Gaussian Splatting (c.f. Section 4.1, Section D.1), as well as black-box attacks for Scaffold-GS (c.f. Section 4.2, Section D.2).
>
> To further test the effectiveness of our approach, we propose conducting experiments on Mip-Splatting [1], which received the best student paper award at CVPR 2024 and has been highly popular (with 178 citations as of November 20th, 2024). Mip-Splatting is an advanced variant of the original 3D Gaussian Splatting, incorporating a 3D smoothing filter and a 2D Mip filter. These enhancements help eliminate various artifacts and achieve alias-free renderings. Part of results are shown in the following table.
>
> *Constrained Black-box attack agaisnt Mip-Splatting*
>
> | Scene        | Number of Gaussians (poisoned) | GPU Memory (poisoned)   | Training Time (poisoned) |
> |:--:|:--:|:--:|:--:|
> | eps16-NS-chair  | 1.088 M (2.92x)               | 33042 MB (5.23x)        | 13.75 min (1.75x)    |
> | eps16-NS-hotdog | 1.437 M (7.18x)               | 58878 MB (11.01x)       | 20.08 min (2.84x)        |
> | eps16-NS-lego   | 1.127 M (3.67x)               | 48682 MB (7.27x)        | 16.43 min (2.26x)     |
> | eps16-NS-ship   | 1.793 M (5.43x)               | 61026 MB (9.39x)        | 21.05 min (2.41x)        |
> | eps16-MIP-bicycle  | **DoS**   | **DoS**  | **DoS**     |
> | eps16-MIP-bonsai   | 13.016 M (7.79x)              | 80826 MB (2.62x)        | 61.82 min (2.61x)        |
> | eps16-MIP-counter  | 8.329 M (5.58x)               | 79904 MB (4.10x)        | 56.75 min (2.16x)        |
> | eps16-MIP-room     | 12.949 M (6.23x)              | 81130 MB (2.57x)        | 77.83 min (2.87x)        |
>
>
> *Unconstrained Black-box Poison-splat attack  against Mip-Splatting*
>
> | Scene              | Number of Gaussians (poisoned) | GPU Memory (poisoned)   | Training Time (poisoned) |
> |:--:|:--:|:--:|:--:|
> | unconstrained-NS-chair | 6.106 M (16.41x)             | 80732 MB (12.78x)       | 57.73 min (7.35x)        |
> | unconstrained-NS-hotdog| 6.876 M (34.38x)             | 80630 MB (15.07x)       | 59.80 min (8.45x)        |
> | unconstrained-NS-lego  | 6.472 M (21.08x)             | 80668 MB (12.05x)       | 64.93 min (8.92x)        |
> | unconstrained-NS-ship  | 6.848 M (20.75x)             | 81236 MB (12.50x)       | 66.15 min (7.56x)        |
> | unconstrained-MIP-bicycle  | **DoS**   | **DoS**  | **DoS**   |
> | unconstrained-MIP-bonsai   | **DoS**   | **DoS**  | **DoS**   |
> | unconstrained-MIP-counter  | **DoS**   | **DoS**  | **DoS**   |
> | unconstrained-MIP-room     | **DoS**   | **DoS**  | **DoS**   |
>
> We found that Mip-Splatting consumes more GPU memory compared to the original 3D Gaussian Splatting, making it more prone to the worst  attack consequence of running Out-of-Memory. As illustrated in above table, even when the attack perturbation is constrained to $\epsilon=16/255$, the GPU memory consumption nearly reached the 80 GB  capacity of Nvidia A800. When we apply an unconstrained attack, all scenes in the MIP-NeRF360 dataset will result in denial-of-service.
>
> We examined the code implementation of Mip-Splatting (https://github.com/autonomousvision/mip-splatting), and identified that it uses quantile computation in its Gaussian model densification function (In Line 524 of `mip-splatting/scene/gaussian_model.py`) which requires massive GPU memory and  easily  triggers out-of-memory. Our attack highlights the vulnerability in various 3DGS algorithm implementations.
>
> **We put the full result of black-box attack on Mip-Splatting in Appendix Section D.2 and Table 6.**
>
> ***Reference***
>
> *[1] Yu, Zehao, et al. "Mip-splatting: Alias-free 3d gaussian splatting." Proceedings of the IEEE/CVF Conference on Computer Vision and Pattern Recognition. 2024.*

---

> ### Author Response · Authors · 2024-11-22
> **Response to Reviewer THFm (Split 5 of 5)**
>
> **Q6(1). Worst consequences of the attack. Prior knowledge for victim computation resource.**
> >`Q6(1). The primary objective of the attack is to consume excessive computational resources, but what are the most serious consequences of this? Is it a denial-of-service (DoS) scenario? Additionally, how difficult would it be to achieve significant damage? For instance, if an attacker aims to prevent a model from completing its training, would this require prior knowledge of the service provider's computational resources?`
>
> Yes, the most severe consequence of the attack would be a service crash (i.e. denial-of-service).
>
> We do not assume the attacker has the prior knowledge of victim's computational resources. The attack is hardware-agnostic, and attacker only aims to increase computation cost as much as possible. We conduct experiments on A800 since they are largest memory GPUs we can access. Referring to **Q4 of Reviewer SqKM**, if victim uses a common GPU with smaller memory (e.g. A5000 or RTX 4090), it is quite easy to trigger an out-of-memory error (i.e. denial-of-service), as illustrated in the table below.
>
> |Scene - $\epsilon= 16/255$ | A800 | RTX 4090 | RTX A5000 |
> |:--:|:--:|:--:|:--:|
> | NS-hotdog | 29747 MB / 17.43 min | **OOM** | **OOM**|
> | NS-lego | 9960 MB / 13.02 min | 9413 MB / 7.88 min | 10499 MB / 15.73 min |
> | NS-materials | 4886 MB / 9.92 min| 3659 MB / 4.80 min | 5034 MB / 13.33 min
> | NS-ship | 16666 MB / 16.12 min| 15339 MB / 9.82 min | 17580 MB / 20.50 min |
> | MIP-bicycle| 27074 MB / 44.44 min | **OOM** | **OOM**|
> | MIP-bonsai | 20564 MB / 32.51 min | 18853 MB / 28.97 min | 21670 MB / 47.11 min |
> | MIP-counter| 29796 MB / 37.63 min |**OOM** | **OOM**|
> | MIP-garden | 23115 MB / 40.82 min | 21297 MB / 39.72 min | **OOM** |
>
> **Q6(2).  Assessment of attack success or failure.**
> >`Q6(2) Lastly, it is unclear how the success or failure of this type of attack should be assessed. For example, if the service provider’s computational resources are sufficient to handle the increased demand, should the attack then be considered unsuccessful?`
>
> Following the previous question and our response to your **W2**, we consider an attack is more successful if it can introduce more extra computation cost due to the poison perturbation. Even if the attack does not cause the worst consequence which is denial-of-service, we can say such attack is successful if it over-consumes resources and degrades service quality for legitimate users.
>
> For example, our unconstrained attacks on NeRF-Synthetic result in **17.29 times more GPU memory** and **4.77 times more training time** on average compared with unattacked. More clearly put, the computation resource for **training one time on our poisoned datasets** will consume resources enough for **82.47 times of normal training**.

---

> ### Author Response · Authors · 2024-11-26
>
> Dear Reviewer THFm,
>
> Thank you for your time and effort in reviewing our paper. Given the limited time for discussion, we would appreciate it if you could let us know whether we have fully addressed your concerns; or if there are any further questions, we are happy to help resolve.
>
> Best regards,
>
> Author of #968

---

> ### Comment · Reviewer_THFm · 2024-11-27
>
> I am appreciative of the authors' detailed response and the effort they put into providing new results. I am satisfied with their response and have improved my rating.

---

> > ### Author Response · Authors · 2024-11-27
> >
> > We are thrilled to receive your acknowledgment of our work and rebuttal. Your questions have opened up new perspectives for us and helped us think more comprehensively. Thank you once again for your recognition, which is highly valuable and incredibly encouraging to us!

---

### Official Review · Reviewer_gaGQ · 2024-11-03

**Soundness:** 3
**Presentation:** 3
**Contribution:** 3
**Rating:** 6
**Confidence:** 4

**Summary:**

The paper discovers a security vulnerability in 3DGS. It shows that the computation cost of
training 3DGS could be maliciously tampered by poisoning the input data. An attack named Poison-splat is presented.

I have read the response of the authors and the comments of other reviewers. I would recommend weak accept.

**Strengths:**

1. The paper is well written and organized.
2. The paper reveals that the flexibility in model complexity of 3DGS can become a security backdoor, making it vulnerable to computation cost attack.
3. Attacks are formulated and extensive experiments are conducted.

**Weaknesses:**

1. Is the attack practically feasible in real-world scenarios, or is it only feasible in theory?
2. In the work, the authors approximate the outer maximum objective with the number of Gaussians, which appears to be a theoretical assumption that may not apply in real-world scenarios.

**Questions:**

My concerns mainly lie in the practically feasible of the proposed attack.
1. Is the attack practically feasible in real-world scenarios, or is it only feasible in theory?
2. In the work, the authors approximate the outer maximum objective with the number of Gaussians, which appears to be a theoretical assumption that may not apply in real-world scenarios.

---

> ### Author Response · Authors · 2024-11-22
> **Response to Reviewer gaGQ**
>
> Thanks for your effort in reviewing our paper, and we glad the reviewer recognized our paper is well written and organized, and our experiments are extensive. We respond to reviewer's concern as follows.
>
> **W1&Q1. Feasibility in real world.**
> >`W1&Q1. Is the attack practically feasible in real-world scenarios, or is it only feasible in theory?`
>
> Thank you for paying attention to the real-world feasibility. In our attack formulation, we don't impose too much assumptions on the victim 3DGS trainer, and the attack succeeds on various content of data (including MIP-NeRF360 and Tanks-and-Temples which are real-world scene captures). Even when the victim algorithm details are unknown, attack still succeeds and even frequently triggers out-of-memory error (c.f. **Q5 of Reviewer THFm**).
>
> The practicality of our attack is further demonstrated by our tests on the real-world online 3DGS service platform, Polycam(https://poly.cam/tools/gaussian-splatting). Polycam supports user to upload 20-100 images captured from various angles to create a Gaussian Splatting 3D model. We upload the poisoned dataset we made using the proxy model, treating Polycam as a real-world black-box model, unaware of the underlying GPU device, nor the specific 3DGS algorithm variant Polycam is running. We can infer three information through the website interface: (1) whether training is successful (2) service responding time (3) the file size of the downloadable reconstructed 3DGS model.
>
> We conducted tests of our attack using `room` scene from MIP-NeRF360. The scene has a total of 311 views, but since Polycam only supports between 20 to 100 views, we uploaded the first $v$ number of views for reconstruction and keep the uploaded views uniform for clean and poisoned cases. The service responses are recorded in the table below:
>
> | Number of Views | Clean file size / response time | Poisoned file size / response time |
> |:---:|:---:|:---:|
> |20 | 15.8 MB / 16 min | 31.5 MB / 24 min|
> |60 | 29.0 MB / 29 min | 46.1 MB / 56 min|
> |70 | 32.4 MB / 33 min  |**Service failed** |
> |100| 40.2 MB / 45 min | **Service failed** |
>
> Although not knowing specific algorithm variant or running device used by Polycam, our attack successfully results in a more complex 3D model and extends training time. As input views cover more of the scene, both 3D model file size and training time increase; notably, when the number of views reached 70, while clean dataset still receive a response, out poisoned dataset encounter a service failure. Although we don't have direct access to memory consumption of the online service, the increasing trend in file size provides sufficient reason to believe that our poisoning exceeds the computation resource budget of a single reconstruction service. This evidence highlights the effectiveness of our Poison-splat attack in real-world scenarios.
>
> **W2&Q2. Outer objective approximation seems a theoretical assumption.**
> >`W2&Q2. In the work, the authors approximate the outer maximum objective with the number of Gaussians, which appears to be a theoretical assumption that may not apply in real-world scenarios.`
>
> Sorry we might leave potential confusions. Our approximation is based on the fact that computation cost has strong positive correlation with number of Gaussians (c.f Figure 2(a) and Figure 2(b)). Our experimental results (c.f. Section 4) and our testing on real-world 3DGS platforms Polycam (refer to **your Q1**) fully supports this claim.

---

> ### Author Response · Authors · 2024-11-26
>
> Dear Reviewer gaGQ,
>
> Thank you for your time and effort in reviewing our paper. Given the limited time for discussion, we would appreciate it if you could let us know whether we have fully addressed your concerns; or if there are any further questions, we are happy to help resolve.
>
> Best regards,
>
> Author of #968

---

### Official Review · Reviewer_ybEK · 2024-11-03

**Soundness:** 3
**Presentation:** 3
**Contribution:** 3
**Rating:** 8
**Confidence:** 4

**Summary:**

This paper proposes an adversarial attack against 3D Gaussian splatting, aiming at increasing the computational cost of this process. Their attack is based on the flexibility of this algorithm, in which the computational cost will change dynamically according to the input image features. Their attack named poison-splat leverages a proxy 3DGS model and the improvement of the total variation score to increase the number of gaussians required in computation, hence bring a huge computational cost regarding GPU memory usage and training time. Their evaluation has included both white-box and black-box attack results and discussed simple defense strategies.

**Strengths:**

1. This work identifies a new kind of vulnerability in 3DGS systems, which is the computational cost attack.
2. Authors have proposed an efficient algorithm to optimize a perturbation to increase the number of gaussians required in 3DGS.
3. The presentation of the paper is clear and easy to follow.
4. Evaluation results demonstrate the good attack performance in both black-box and white-box settings.

**Weaknesses:**

1. The constraint of the perturbation (epsilon = 16/255) seems large, and the quality of the resulted image could be affected. More ablation studies may be conducted to evaluate other constraint thresholds.
2. A simple defense might be smoothing the input images before conducting 3DGS, which seems an adaptive defense regarding your perturbations to the input. You may discuss or evaluate the effectiveness and negative impact of such defense.

**Questions:**

1. Since there are many online services using 3DGS, as you mentioned in the paper, have you evaluated the real-world attack performance of your technique on those application? Will the responding time be extended or causing deny of service?

---

> ### Author Response · Authors · 2024-11-22
> **Response to Reviewer ybEK (split 1 of 2)**
>
> Thanks for the effort in reviewing our paper, we are glad that our novelty, efficiency, presentation and good performance are recognized. We address your questions as follows.
>
> **W1. Other constraint thresholds.**
> >`W1. The constraint of the perturbation (epsilon = 16/255) seems large, and the quality of the resulted image could be affected. More ablation studies may be conducted to evaluate other constraint thresholds.`
>
> Thank you for raising this point. To evaluate the trade-off between attack performance and image quality, we further conducted experiments of different perturbation constraints $\epsilon=8/255$ and $\epsilon=24/255$. It is evident that a larger perturbation range boosts the attack performance, consuming more GPU memory and longer training time. However, it also results in higher image distortion, as indicated by lower SSIM and PSNR values compared with clean images.
>
> We show one scene from each of NeRF-Synthetic and MIP-NeRF360:
>
>
> | Attack setting   | Number of Gaussians | GPU memory | Training time | SSIM | PSNR |
> |------------------|:---:|:---:|:---:|:---:|:---:|
> | **Nerf-Synthetic-ship**         |                     |            |               |      |      |
> | clean            | 0.272 M            | 3692 MB    | 8.87 min      | -    | -    |
> | $\epsilon=8/255$ | 0.516 M            | 5574 MB    | 11.01 min     | 0.55 | 33.42 |
> | $\epsilon=16/255$| 1.071 M            | 16666 MB   | 16.12 min     | 0.33 | 27.50 |
> | $\epsilon=24/255$| 1.365 M            | 29828 MB   | 18.46 min     | 0.24 | 24.03 |
> | unconstrained    | 4.317 M            | 80956 MB   | 44.11 min     | 0.04 | 5.31 |
> | **MIP-NeRF360-counter**         |                     |            |               |      |      |
> | clean            | 1.195 M            | 10750 MB    | 26.46 min      | -    | -    |
> | $\epsilon=8/255$ | 1.739 M            | 15133 MB    | 28.06 min     | 0.80 | 32.24 |
> | $\epsilon=16/255$| 4.628 M             | 29796 MB   | 37.63 min     | 0.52 | 26.78 |
> | $\epsilon=24/255$| 6.649 M             | 47607 MB   | 43.68 min     | 0.35 | 23.45 |
> | unconstrained    | 11.167 M            | 80732 MB   | 62.04 min     | 0.01 | 6.64 |
>
> Please refer to the **full result in Appendix Section G, Table 10 and Table 11** in our revised paper.
>
> ---
> **W2. Smoothing the input images can be a defense.**
> >`W2. A simple defense might be smoothing the input images before conducting 3DGS, which seems an adaptive defense regarding your perturbations to the input. You may discuss or evaluate the effectiveness and negative impact of such defense.`
>
> Thank you for this constructive suggestion. We will show that image smoothing as a pre-processing to Gaussian Splatting training procedure is not an ideal defense. Without a reliable detection method, defenders may only resort to universally applying image smoothing to all incoming data. This pre-processing will significantly compromise reconstruction quality.
>
> We found that although image smoothing may reduce GPU consumption to some extent, it severly undermines efforts to preserve fine image details[1][2][3]. As illustrated in **Figure 7 in Appendix Section J**, applying common smoothing techniques such as Gaussian filtering or Bilateral filtering[4] leads to a substantial degradation in reconstruction quality. For instance, on the `chair` scene of NeRF-Synthetic dataset, reconstruction achieves 36.91 dB PSNR without pre-processing; however, with Gaussian or Bilateral filtering, the PSNR drops sharply to around 25 dB. This level of degradation is clearly undesirable. Given these challenges, we urge the community to develop more sophiscated detection and defense mechanisms against computation cost attacks, balancing resource consumption with the preservation of image quality.
>
> *Reference*
>
> *[1] Yu, Zehao, et al. "Mip-splatting: Alias-free 3d gaussian splatting." Proceedings of the IEEE/CVF Conference on Computer Vision and Pattern Recognition. 2024.*
>
> *[2] Ye, Zongxin, et al. "Absgs: Recovering fine details in 3d gaussian splatting." Proceedings of the 32nd ACM International Conference on Multimedia. 2024.*
>
> *[3] Yan, Zhiwen, et al. "Multi-scale 3d gaussian splatting for anti-aliased rendering." Proceedings of the IEEE/CVF Conference on Computer Vision and Pattern Recognition. 2024.*
>
> *[4] Tomasi, Carlo, and Roberto Manduchi. "Bilateral filtering for gray and color images." Sixth international conference on computer vision (IEEE Cat. No. 98CH36271). IEEE, 1998.*

---

> ### Author Response · Authors · 2024-11-22
> **Response to Reviewer ybEK (split 2 of 2)**
>
> **Q1. Evaluate the attack on real-world 3DGS online services.**
> >`Q1. Since there are many online services using 3DGS, as you mentioned in the paper, have you evaluated the real-world attack performance of your technique on those application? Will the responding time be extended or causing deny of service?`
>
> Thank you for this constructive advise. We further evaluated our attack on the platform of a real-world 3DGS online service provider, Polycam (https://poly.cam/tools/gaussian-splatting). Polycam supports user to upload a set of images captured from various angles to create a Gaussian Splatting 3D model.
>
> Polycam supports user to upload 20-100 images for a specific scene. Since it is an online 3DGS service provided for user, we do not know the underlying GPU device, nor the specific 3DGS algorithm variant Polycam is running. We only upload the poisoned dataset we made using the proxy model, treating Polycam as a real-world black-box model. We can infer three information through the website interface: (1) whether training is successful (2) service responding time (3) the file size of the downloadable reconstructed 3DGS model.
>
> We conducted tests of our attack using `room` scene from MIP-NeRF360. The scene has a total of 311 views, but since Polycam only supports between 20 to 100 views, we uploaded the first $v$ number of views for reconstruction and keep the uploaded views uniform for clean and poisoned cases. The service responses are recorded in the table below:
>
>
> | Number of Views | Clean file size / response time | Poisoned file size / response time |
> |:---:|:---:|:---:|
> |20 | 15.8 MB / 16 min | 31.5 MB / 24 min|
> |60 | 29.0 MB / 29 min | 46.1 MB / 56 min|
> |70 | 32.4 MB / 33 min  |**Service failed** |
> |100| 40.2 MB / 45 min | **Service failed** |
>
> Although not knowing specific algorithm variant or running device used by Polycam, our attack successfully results in a more complex 3D model and extends training time. As input views cover more of the scene, both 3D model file size and training time increase; notably, when the number of views reached 70, while clean dataset still receive a response, out poisoned dataset encounter a service failure. Although we don't have direct access to memory consumption of the online service, the increasing trend in file size provides sufficient reason to believe that our poisoning exceeds the computation resource budget of a single reconstruction service. This evidence highlights the effectiveness of our Poison-splat attack in real-world scenarios.

---

> ### Author Response · Authors · 2024-11-26
>
> Dear Reviewer ybEK,
>
> Thank you for your time and effort in reviewing our paper. Given the limited time for discussion, we would appreciate it if you could let us know whether we have fully addressed your concerns; or if there are any further questions, we are happy to help resolve.
>
> Best regards,
>
> Author of #968

---

> ### Comment · Reviewer_ybEK · 2024-11-26
>
> Thanks for your detailed response. All my concerns have been successfully resolved and I have raised my rating. Please update the additional results and discussions to your paper. I believe they could make the work more comprehensive.

---

> > ### Author Response · Authors · 2024-11-26
> >
> > Thank you sincerely for your insightful comments. We will surely include all the additional results and discussions in the camera-ready version. Thank you again for your valuable feedback!

---

### Official Review · Reviewer_SqKM · 2024-11-03

**Soundness:** 3
**Presentation:** 4
**Contribution:** 3
**Rating:** 8
**Confidence:** 3

**Summary:**

The paper introduces Poison-splat, a data poisoning attack targeting the training phase of 3D Gaussian Splatting (3DGS). It exposes a vulnerability in the adaptive model complexity of 3DGS, showing how manipulated input data can significantly escalate computation costs during training, potentially resulting in a Denial-of-Service by consuming all available memory.

**Strengths:**

+ A unique computation cost attack targeting 3D Gaussian Splatting.
+ Highlights practical vulnerabilities in commercial 3D reconstruction services.
+ Thorough experimentation across various datasets.

**Weaknesses:**

- The paper frames the problem as a data poisoning attack. However, it does not clearly elaborate on the poisoning ratio required for Poison-Splat to be effective. Additionally, the implications of varying poisoning ratios, particularly their impact on the stealthiness and overall effectiveness of the attack, are not thoroughly discussed.

- The paper mentions that Poison-Splat maximizes the number of Gaussians by enhancing the sharpness of 3D objects through controlling the smoothness factor. However, it is not clearly explained how the smoothness threshold is defined to ensure an effective attack. Additionally, the impact of this threshold on the stealthiness of the attack remains unclear.

- Algorithm 1 indicates that generating backdoor samples with Poison-Splat requires a quadratic time complexity relative to the number of iterations, which raises concerns about the practicality of this approach during training.

- Would the Poison-Splat technique maintain its effectiveness when the 3DGS algorithm is trained in a multi-GPU environment? Additionally, the attack should be assessed on various GPUs with different clock frequencies and memory bandwidths to evaluate the generalizability of the approach.

- The paper states a basic defense against Poison-Spat by limiting the number of Gaussians, but it does not specify the threshold for the number considered in this defense. It would be beneficial to include an evaluation that explores the effect of varying these Gaussian limits.

- The definition of the threat model for the Poison-Splat attack could be more precise, particularly in specifying attacker capabilities and constraints. Explicitly define white-box and black-box scenarios for the proxy model.

- Why does the attack perform well on specific datasets in the white-box scenario but less effectively on others, such as the Tanks-and-Temples data (as shown in Table 1 and Table 3)? Can authors provide additional reasoning?

**Questions:**

1. What poisoning ratio is needed for Poison-Splat to be effective, and how does it affect stealth and impact?
2. How is the smoothness threshold defined, and how does it impact the stealth of the attack?
3. Does the quadratic time complexity of Algorithm 1 raise practical concerns during training?
4. Is Poison-Splat still effective in multi-GPU settings, and how does it perform across GPUs with different specs?
5. What is the Gaussian limit threshold in the basic defense, and how do varying limits affect the attack?

---

> ### Author Response · Authors · 2024-11-22
> **Response to Reviewer SqKM (split 1 of 4)**
>
> Thanks for the comprehensive and constructive comments, particularly for recognizing our work is practical and thorough. We address all the concerns as follows.
>
> **W1&Q1. Poisoning ratio.**
> >`W1. The paper frames the problem as a data poisoning attack. However, it does not clearly elaborate on the poisoning ratio required for Poison-Splat to be effective. Additionally, the implications of varying poisoning ratios, particularly their impact on the stealthiness and overall effectiveness of the attack, are not thoroughly discussed.`
>
> >`Q1. What poisoning ratio is needed for Poison-Splat to be effective, and how does it affect stealth and impact?`
>
> Thank you for raising this question. In the main paper, we only consider 100% poison rate based on the assumption that attacker can fully control the data collection process, since 3D datasets are typically composed of images of the same scene from different angles.
>
> To answer your question, we further explore how varying poisoning ratio can affect the attack’s effectiveness, as reported in the following table:
> | Scene             | Poison Ratio | Number of Gaussians | GPU memory (MB)  | Training time (min) |
> |-------------------|--------------|---------------------|------------------|---------------------|
> | **NS-hotdog-eps16** | clean        | 0.185 M ± 0.000 M   | 3336 MB ± 11 MB  | 8.57 min ± 0.30 min |
> |                   | 20%          | 0.203 M ± 0.004 M   | 3286 MB ± 70 MB  | 8.84 min ± 0.27 min |
> |                   | 40%          | 0.279 M ± 0.024 M   | 3896 MB ± 286 MB | 9.79 min ± 0.53 min |
> |                   | 60%          | 0.501 M ± 0.054 M   | 7367 MB ± 1200 MB| 11.33 min ± 0.82 min|
> |                   | 80%          | 0.806 M ± 0.018 M   | 13621 MB ± 861 MB| 14.02 min ± 0.60 min|
> |                   | 100%         | 1.147 M ± 0.003 M   | 29747 MB ± 57 MB | 17.43 min ± 0.61 min|
> |----|-----|----------|----------------------|--------------------------|
> | **MIP-counter-eps16**| clean        | 1.195 M ± 0.005 M     | 10750 MB ± 104 MB    | 26.46 min ± 0.57 min     |
> |                      | 20%          | 1.221 M ± 0.005 M     | 11043 MB ± 141 MB    | 27.41 min ± 0.85 min     |
> |                      | 40%          | 1.358 M ± 0.030 M     | 11535 MB ± 147 MB    | 27.97 min ± 0.47 min     |
> |                      | 60%          | 2.005 M ± 0.056 M     | 13167 MB ± 227 MB    | 28.87 min ± 0.52 min     |
> |                      | 80%          | 3.273 M ± 0.119 M     | 19578 MB ± 1417 MB   | 32.99 min ± 0.57 min     |
> |                      | 100%         | 4.628 M ± 0.014 M     | 29796 MB ± 187 MB    | 37.63 min ± 0.48 min     |
>
> It is evident that attack performance, in terms of both GPU memory usage and training time extension, becomes stronger as poisoning rate increases. Here we randomly selected the poisoned views and report the standard deviation across three individual runs. Please refer to **Table 12 in Appendix Section H** to see the full results.
>
> ---
> **W2&Q2. Smoothness threshold and its relationship with attack performance and stealth.**
> >`W2. The paper mentions that Poison-Splat maximizes the number of Gaussians by enhancing the sharpness of 3D objects through controlling the smoothness factor. However, it is not clearly explained how the smoothness threshold is defined to ensure an effective attack. Additionally, the impact of this threshold on the stealthiness of the attack remains unclear.`
>
> >`Q2. How is the smoothness threshold defined, and how does it impact the stealth of the attack?`
>
> Thank you for your question. We notice that the term "smoothness threshold" was not explicitly mentioned in our manuscript, and we speculate that you might be referring to what we termed as the "perturbation range" in our paper.
>
> Generally, as the perturbation range increases, the attack becomes more effective but loses stealthiness due to greater distortion of image. The attacker's choice of perturbation range depends on their goals: if prioritizing attack effectiveness, they may opt for unlimited perturbations. If stealth is more important, they might chooses a smaller $\epsilon$ to make the changes hard to detect.
>
> We conducted an ablation study on the perturbation range $\epsilon$ where the trend of attack effectiveness and stealth in relation to $\epsilon$ is quite apparent. Please refer to **Table 10 and Table 11 in Appendix Section G** for detailed results.

---

> ### Author Response · Authors · 2024-11-22
> **Response to Reviewer SqKM (split 2 of 4)**
>
> ---
> **W3&Q3. Quadratic complexity and practical concern.**
> >`W3. Algorithm 1 indicates that generating backdoor samples with Poison-Splat requires a quadratic time complexity relative to the number of iterations, which raises concerns about the practicality of this approach during training.`
>
> >`Q3. Does the quadratic time complexity of Algorithm 1 raise practical concerns during training?`
>
> According to Algorithm 1, we need to solve a bi-level optimization problem, and the time complexity is related to $T$ (inner iterations) and $\tilde{T}$ (outer iterations). However in practice, our inner iterations $T$ are far fewer than outer iterations $\tilde{T}$, and the actual time cost of attacker is even less than victim's training time, which is an advantage of our attack.
>
> Specifically, the total running time of algorithm 1 is decided by two parts:
>
> 1. Train a proxy model on clean data (Line 2 of Algorithm 1)
> 2. Solve the bi-level optimization (Line 3 to Line 13 of Algorithm 1)
>
> For example, for unconstrained attacks on MIP-NeRF360, we set $T=6000$ and $\tilde{T}=25$, and the time of attack  is totally acceptable, and is even shorter than the victim training, as shown in the table below:
>
> | Scene | [Attacker: Proxy training + Bi-level optimization] | [Victim: Training time] |
> |:----:|:----:|:----:|
> |Bicycle | 33.42 + 16.57 min | 81.48 min |
> |Bonsai  | 21.68 + 14.00 min | 75.18 min |
> |Counter | 26.46 + 15.23 min | 62.04 min |
> |Flowers | 25.47 + 15.05 min | 62.62 min |
> |Garden  | 33.19 + 15.07 min | 83.81 min |
> |Kitchen | 26.61 + 14.77 min | 73.04 min |
> |Room    | 25.36 + 16.50 min | 76.25 min |
> |Stump   | 27.06 + 15.63 min | 51.51 min |
>
> Lastly, the primary motivation of our work is to expose the severity of this security backdoor, and attack efficiency is beyond the scope of this paper. We hope to leave the improvement of attack efficiency in future studies.
>
> ---
> **W4&Q4(1). Would attack maintain effective in multi-GPU environment?**
> >`W4. Would the Poison-Splat technique maintain its effectiveness when the 3DGS algorithm is trained in a multi-GPU environment? `
>
> >`Q4. Is Poison-Splat still effective in multi-GPU settings?`
>
> Our attack is hardware-agnostic, which does not assume the hardward platform victim is using. The computation resource consumption is decided by number of Gaussians. Even if 3DGS training is carried out in a multi-GPU environment, as long as the total number of Gaussians remains at the same level, Poison-Splat attack will still maintain effective.
>
> **W4&Q4(2). Attack should be tested on various GPUs with different clock frequencies and memory bandwidths.**
> >`W4. Additionally, the attack should be assessed on various GPUs with different clock frequencies and memory bandwidths to evaluate the generalizability of the approach.`
>
> >`Q4. how does it perform across GPUs with different specs?`
>
> In our paper, we conducted our attack tests using an 80GB-A100 GPU, the largest memory card available to us, to benchmark the extreme resource consumption of our attack.  Following the reviewer's advice, we test our attack on another two different computation cards A5000 and RTX 4090. The detailed comparison of GPU specifications:
>
> | GPU type | Memory Size | Memory Bandwidth | Clock Speed |
> |:------:|:---: | :----:|:------:|
> |A800 | 80 GB | 2.04 TB/s | 1155 MHz |
> |RTX 4090 | 24 GB | 1.01 TB/s | 2235 MHz |
> |RTX A5000 | 24 GB | 768 GB/s | 1170 MHz |
>
> The benchmark results (GPU memory / training time) of our poisoned dataset on these computation cards are presented in the table below. (**OOM** stands for out-of-memory error, which effectively indicates denial-of-service.)
>
> |Scene - $\epsilon= 16/255$ |  A800 | RTX 4090 | RTX A5000 |
> |:------:|:---: | :---:|:----:|
> | NS-hotdog | 29747 MB / 17.43 min | **OOM** | **OOM**|
> | NS-lego | 9960 MB / 13.02 min | 9413 MB / 7.88 min | 10499 MB / 15.73 min |
> | NS-materials | 4886 MB / 9.92 min| 3659 MB / 4.80 min | 5034 MB / 13.33 min
> | NS-ship | 16666 MB / 16.12 min| 15339 MB / 9.82 min | 17580 MB / 20.50 min |
> | MIP-bicycle| 27074 MB / 44.44 min | **OOM** | **OOM**|
> | MIP-bonsai | 20564 MB / 32.51 min | 18853 MB / 28.97 min | 21670 MB / 47.11 min |
> | MIP-counter| 29796 MB / 37.63 min |**OOM** | **OOM**|
> | MIP-garden | 23115 MB / 40.82 min | 21297 MB / 39.72 min | **OOM** |
>
> Due to variations in specifications and underlying hardware architectures, the same poisoned dataset can lead to different GPU memory usage and run times on different devices. It is evident that computation cards with less memory are particularly vulnerable to the most severe consequence of our attack:  out-of-memory and service disruption, even when only a constrained attack is applied. If an unconstrained attack is applied, it would result in a denial-of-service across all scenarios on these 24GB GPUs (c.f. Table 4 in Appendix Section D.1).

---

> ### Author Response · Authors · 2024-11-22
> **Response to Reviewer SqKM (split 3 of 4)**
>
> **W5&Q5. Defense threshold of limiting Gaussian numbers.**
> >`W5. The paper states a basic defense against Poison-Spat by limiting the number of Gaussians, but it does not specify the threshold for the number considered in this defense. It would be beneficial to include an evaluation that explores the effect of varying these Gaussian limits.`
>
> >`Q5. What is the Gaussian limit threshold in the basic defense, and how do varying limits affect the attack?`
>
> Thank you for raising this question. In the paper we set the defense threshold to ensure a tight wrapping of necessary Gaussians on the clean scene. More specifically, for three scenes we show in Figure 5, we set defense threshold as follows:
>
> | Scene | #Gaussians on clean data | #Gaussians as defense threshold |
> |:---:|:---:|:---:|
> |Bicycle|5.793 Million|6 Million|
> |Flowers|3.508 Million|4 Million|
> |Room|1.513 Million|2 Million|
>
> Setting a uniform defense threshold for Gaussian number is challenging due to the significant variance in different scenes. A low threshold can safeguard the victim’s computational resources from over-consumption, but may substantially degrade the reconstruction quality for complex scenes. Choosing the defense threshold should consider this resource-quality tradeoff.  Taking `room` of Nerf-Synthetic as an example, we further tested the defense threshold across 2 million to 7 million, as shown in the following table:
>
> | Defense threshold | Max GPU memory(MB) | Reconstruction PSNR(dB) |
> |:---:|:----:|:---:|
> | *Clean input* |12316|29.21|
> |2 Million |14192|19.03|
> |3 Million |16472|22.27|
> |4 Million |18784|24.11|
> |5 Million |23642|24.49|
> |6 Million |36214|28.02|
> |Poison + No Defense|46238|29.08|
>
> As the defense threshold becomes tighter, the maximum GPU memory usage is confined within a safer range. However, this will cause a significant drop in the reconstruction’s PSNR, falling to as low as 19.03, which nearly makes it unusable. (c.f. Figure 6 (A)). **We add a new section I in Appendix, along with  visualizations of different defense threshold in Figure 6.** Please refer to our revised paper.
>
> ---
> **W6. Precise definition of threat model.**
> >`W6. The definition of the threat model for the Poison-Splat attack could be more precise, particularly in specifying attacker capabilities and constraints. Explicitly define white-box and black-box scenarios for the proxy model.`
>
>
> Thank you for the advice. We have put another section (**Appendix Section C**) in our revised paper to provide more clarity of the threat model.
>
> Throughout the paper, we assume the attacker has following goal, input, output, information and constraints:
>
> **Attacker Goal**: To increase the computation cost and over-consume computation resource of 3DGS service provider.
>
> **Attacker Input**: Clean data.
>
> **Attacker Output**: Poisoned data by running attack on the clean data.
>
> **Attacker Information**:  Attacker doesn't need to know the underlying hardware device of the victim.
>
>    - White-box: Attacker knows the specific 3DGS configuration of the victim.
>    - Black-box: Attacker only knows victim is using 3DGS, but does not know which variant of 3DGS representation and what configuration the victim is using.
>
> **Attacker Constraint**: Attacker can optionally choose to constrain the perturbation range $\epsilon$ of his additive perturbation to add on clean data.
>
> **Attacker Algorithm**: Algorithm 1.

---

> ### Author Response · Authors · 2024-11-22
> **Response to Reviewer SqKM (split 4 of 4)**
>
> ---
> **W7. Why attack perform less effective on Tanks-and-Temples?**
> >`W7. Why does the attack perform well on specific datasets in the white-box scenario but less effectively on others, such as the Tanks-and-Temples data (as shown in Table 1 and Table 3)? Can authors provide additional reasoning?`
>
> Thank you for paying attention to this performance difference. We also noticed that Poison-Splat attack appears to be more effective on NeRF-Synthetic and MIP-NeRF360 compared to Tanks-and-Temples dataset. We propose a conjecture to explain this based on our observations and experience.
>
> We noticed a big difference in the camera setups across these datasets. Specifically, camera poses in Tanks-and-Temples are more zoomed-in and close-up, while other two datasets are wider, panoramic views. This difference may influence the extent of object surfaces exposed to different camera angles. Intuitively, the more angles from which an object is viewed, the greater the opportunity for optimizing the Total Variation (TV) score. This is due to our bi-level optimization process where, in each iteration, a new camera view is sampled (Line 4 of Algorithm 1), and its TV score is optimized (Line 8 of Algorithm 1). Consequently, greater exposure of an object within the scene leads to more TV scores growth.
>
> Our empirical tests further support this conjecture. We found that more panoramic NeRF-Synthetic and MIP-NeRF360 datasets offers more opportunity to optimize TV scores during poisoning compared to Tanks-and-Temples, leading to higher TV score growth, and more effective attack performance.
>
> | Dataset | TV Score Growth| #Gaussians Growth| GPU Memory Growth | Training Time Growth|
> |:---:|:---:|:---:|:---:|:----:|
> | NeRF-Synthetic | 4.36x | 2.68x | 3.13x | 1.52x|
> | MIP-NeRF360  | 3.18x | 2.81x | 1.97x | 1.43x|
> | Tanks-and-Temples | 2.59x | 1.71x | 1.35x | 1.27x|

---

> ### Author Response · Authors · 2024-11-26
>
> Dear Reviewer SqKM,
>
> Thank you for your time and effort in reviewing our paper. Given the limited time for discussion, we would appreciate it if you could let us know whether we have fully addressed your concerns; or if there are any further questions, we are happy to help resolve.
>
> Best regards,
>
> Author of #968

---

> > ### Comment · Reviewer_SqKM · 2024-11-26
> >
> > I appreciate the authors' detailed response and the effort they invested in providing new results. I am satisfied with their reply and have raised my rating.

---

> > > ### Author Response · Authors · 2024-11-26
> > >
> > > We sincerely appreciate your constructive and detailed feedback, which has greatly contributed to improving the quality of our paper. Your recognition of our efforts is highly encouraging, and we are delighted to have addressed your concerns to your satisfaction. Thank you once again for your valuable suggestions and thoughtful review.

---

### Meta-Review · Area_Chair_NVSV · 2024-12-19

**Metareview:**

This paper proposed a new security vulnerability in 3D Gaussian Splatting. All reviewers provides positive feedback of this paper. AC read all reviewers and rebuttal and recommend this paper as a spotlight paper. This paper also show the practical usage of this attack which is also very important and good. AC hopes the authors can revised the paper based on reviewers' comments.

**Additional Comments On Reviewer Discussion:**

Before rebuttal, reviewers have concerns about key contribution, missing experiments and ablation studies, missing computational cost,  the attack performances on real-world 3DGS online services and so on. Authors did a great job by conducting a large amount of experiments to address reviewers' concerns. Finally, almost all reviewers have increased their scores.  AC hope authors to reorganized the paper to add all these important experiments in the revised version.

---

### Decision · Program_Chairs · 2025-01-22

Accept (Spotlight)